# Reported transgenerational responses to *Pseudomonas aeruginosa* in *Caenorhabditis elegans* are not robust

Daniel Patrick Gainey, Andrey V Shubin, Craig P Hunter*

Department of Molecular and Cellular Biology, Harvard University, Divinity Avenue, The Biological Laboratory, Cambridge, United States

## eLife Assessment

This **important** study reports numerous attempts to replicate reports on transgenerational inheritance of a learned behavior – pathogen avoidance – in *C. elegans*. While the authors observe parental effects that are limited to a single generation (also called intergenerational inheritance), the authors failed to find evidence for transmission over multiple generations, or transgenerational inheritance. The experiments presented are meticulously described, making for **compelling** evidence that in the authors' hands transgenerational inheritance cannot be observed. There remains the possibility that different assay setups explain the failure to reproduce previous observations, although the authors present data suggesting that details of the assay are not that significant. There also remains the possibility that differences in culture conditions or lab environment explain the failure to reproduce previous observations, with updates to the paper having further reduced the probability that this applies here. Even if this were the case, it would imply that the original experimental paradigm was dependent on a very specific context. Given the prominence of the original reports of transgenerational inheritance, the present study is of broad interest to anyone studying genetics, epigenetics, or learned behavior.
[As also pointed out by the authors of this study, the authors of the original reports have provided a response on bioRxiv (DOI: https://doi.org/10.1101/2025.01.21.634111).]

**Abstract** We report our attempt to replicate reports of transgenerational epigenetic inheritance in *Caenorhabditis elegans*. Multiple laboratories report that *C. elegans* adults and their F1 embryos exposed to the pathogen *Pseudomonas aeruginosa* show pathogen aversion behavior and increased *daf-7/TGFβ* reporter gene expression. However, results from one group show persistence of both through the F4 generation. We failed to consistently detect either the avoidance response or elevated *daf-7* expression beyond the F1 generation. We confirmed that the dsRNA transport proteins SID-1 and SID-2 are required for intergenerational (F1) inheritance of pathogen avoidance, but not for the F1 inheritance of elevated *daf-7* expression. Reanalysis of RNA seq data provides additional evidence that this intergenerational inherited PA14 response may be mediated by small RNAs. The experimental methods are well-described, the source materials are readily available, including samples from the reporting laboratory, and we explored a variety of environmental conditions likely to account for lab-to-lab variability. None of these adjustments altered our results. We conclude that this example of transgenerational inheritance lacks robustness, confirm that the intergenerational avoidance response, but not the elevated *daf-7p::gfp* expression in F1 progeny, requires *sid-1* and *sid-2*, and identify candidate siRNAs and target genes that may mediate this intergenerational response.

*For correspondence:
hunter@mcb.harvard.edu

Competing interest: The authors declare that no competing interests exist.

## Introduction

*Caenorhabditis elegans* worms exposed to the pathogen *Pseudomonas aeruginosa* (strain PA14) learn to avoid this specific pathogen upon subsequent exposure (*Zhang et al., 2005*). PA14 exposure also induces the expression of a reporter (*daf-7p::gfp*) for the cytokine TGF-β in the ASI neurons of PA14-exposed animals (*Meisel et al., 2014*). In *Moore et al., 2019* and three follow-up papers (*Kaletsky et al., 2020*; *Moore et al., 2021a*; *Sengupta et al., 2024*), Murphy and colleagues reported that *C. elegans* adults trained to avoid PA14 transmit this learned avoidance and elevated *daf-7p::gfp* expression in ASI neurons to four generations of progeny (F1-F4). They further reported that numerous RNA interference (RNAi) factors, including the dsRNA transport proteins SID-1 and SID-2 (*Winston et al., 2002*; *Feinberg and Hunter, 2003*; *Winston et al., 2007*; *McEwan et al., 2012*), are required for the inheritance of the behavioral response and elevated *daf-7p::gfp* expression (*Kaletsky et al., 2020*; *Moore et al., 2019*). To date, no follow-up reports by independent groups have been published on the transgenerational character of this response. We were keen to reproduce these results and investigate in detail the contributions of the dsRNA transport proteins SID-1 and SID-2 to transgenerational epigenetic inheritance (TEI).

We readily reproduced learned behavior and elevated *daf-7p::gfp* expression in trained parents (P0) and their F1 progeny, but after many attempts and numerous protocol adjustments, we failed to reproducibly replicate inheritance among F2 progeny. While we have been unable to identify a specific methodological cause for our different results, we conclude that this example of TEI is insufficiently robust for experimental investigation.

## Results and discussion

### The reported PA14 training conditions failed to produce an avoidance response or elevated *daf-7p::gfp* expression in ASI neurons among F2 progeny

PA14-exposed animals (P0) and their progeny (F1) avoid PA14 in subsequent choice assays and show elevated *daf-7p::gfp* expression in ASI neurons (*Zhang et al., 2005*; *Moore et al., 2019*; *Kaletsky et al., 2020*; *Pereira et al., 2020*; *Sengupta et al., 2024*). The Murphy group has reported that both responses are transgenerationally inherited by F2-F4 generation worms (*Moore et al., 2019*; *Kaletsky et al., 2020*; *Moore et al., 2021a*; *Sengupta et al., 2024*). Assays for transgenerational inheritance in *C. elegans*, like the choice assay described in *Moore et al., 2019*, are frequently based on the collective behavior of a population. In contrast, the heritable increase in *daf-7p::gfp* expression in ASI neurons is a single animal assay that can be reliably scored in a few dozen animals (*Moore et al., 2019*). Recognizing the potential of this single-animal assay to aid investigation of the reported RNAi-pathway-dependent response to PA14 exposure, we initially attempted to replicate the reported transgenerational *daf-7p::gfp* expression results in the wild-type reporter strain. When these experiments failed to produce an F2 response, we then included the population-based choice assays to assist in troubleshooting.

We performed the choice assay and *daf-7p::gfp* expression assay experiments as described in the published protocols (*Moore et al., 2019*; *Kaletsky et al., 2020*; *Moore et al., 2021b*) with minor adjustments (see Methods and *Supplementary file 1*). The pathogen avoidance response in the trained animals (P0 generation) was robust and often detected among their F1 progeny fertilized during parental exposure (*Figure 1A–C*). However, the transgenerational (F2) response was not detected (*Figure 1A–C*). In contrast, while the magnitude and significance of the elevated *daf-7p::gfp* expression in the ASI neurons in the P0 generation was variable, the response in the F1 generation was strong and statistically highly significant (*Figure 1D–I*). Unlike the results reported in *Moore et al., 2019* and *Kaletsky et al., 2020*, we did not observe a *daf-7p::gfp* response in the F2 generation (*Figure 1D–I*).

Motivated to reproduce the reported results, we obtained and tested independent isolates of the bacterial strains PA14 and OP50. We obtained and used a PA14 isolate from the Balskus lab (Harvard University) and both PA14 and OP50 isolates from the Murphy lab (Princeton University). Similar results were obtained using these reagents (*Figure 1*, *Tables 1 and 2*).

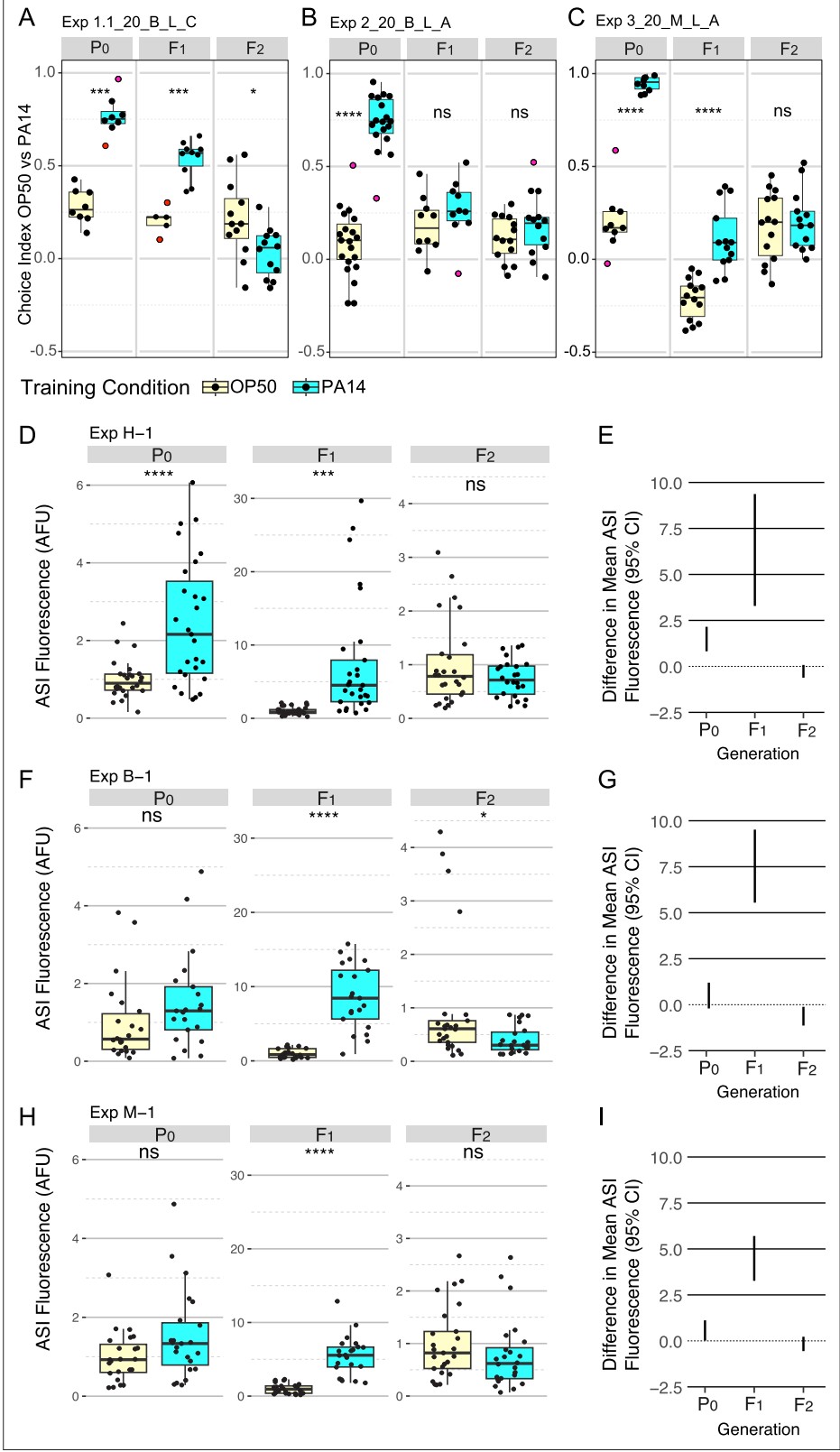

**Figure 1.** P0, F1, and F2 generation responses to P0 PA14 exposure. Three representative experimental results for trained and inherited aversion behavior (**A–C**) and induced and inherited elevated *daf-7p::gfp* in ASI neurons (**D–I**). (**A-C**) Quartile box plots for each generation and training condition showing the distribution of choice index values ([number of animals choosing OP50 - number of animals choosing PA14] / total number of choices) for each

*Figure 1 continued on next page*

*Figure 1 continued*

OP50 vs PA14 choice assay plate following the published protocols. The experimental and assay conditions are indicated in the panel titles; [Exp #] [growth temperature (20 °C or 25 °C)] PA14 isolate (B, Balskus or M, Murphy) light/dark assay condition (L or D) azide/cold paralytic (A or C). (**D, F, H**) Quartile box plots displaying average ASI neuron *daf-7p::gfp* expression levels per worm and (**E, G, I**) show the 95% confidence intervals for the difference in absolute mean between the conditions. For these experiments, the FK181 strain (integrated multi-copy [MC] *daf-7p::gfp* reporter) was cultured at 20 °C at all generations and was exposed to one of three different PA14 isolates (H, Hunter; B, Balskus; M, Murphy labs). AFU arbitrary fluorescence units normalized to mean OP50 levels. Red dots in choice assay results indicate outlier data points that were included in all statistical tests. Statistical significance **** p<0.0001, ***<0.001, **<0.01, *<0.05, ns >0.05. See Methods section for statistical methods.

The online version of this article includes the following source data and figure supplement(s) for figure 1:

**Figure supplement 1.** P0 generation response to PA14 exposure.

**Figure supplement 1—source data 1.** This file contains data on the age of P0 animals at initiation of training.

**Figure supplement 2.** The pathogenicity of PA14 isolates.

**Figure supplement 2—source data 1.** This file contains data on PA14 pathogenesis.

**Figure supplement 3.** Comparison of paralytic treatments.

## Modifying training and growth conditions did not result in more reliable detection of transgenerational responses to PA14 exposure

To attempt to replicate these key observations, we then explored procedural and environmental variations. To train animals to recognize and avoid PA14, gravid wild-type (N2) adults grown on High Growth (HG) plates seeded with *E. coli* strain OP50 were treated with sodium hypochlorite (bleach) to prepare aseptic P0 embryos, which were hatched and grown on HG OP50 plates at 20 °C for 48–52 hr. Larval stage 4 (L4) animals were then gently washed from these plates and placed on Normal Growth (NG) plates seeded with either OP50 (control) or PA14 (training) for 24 hr and then similarly recovered for testing and bleach isolation of F1 embryos. Because the 24 hr training period on PA14 limits reproduction and therefore recovery of F1 progeny, the published protocols (*Moore et al., 2019*; *Moore et al., 2021b*) suggest plating four times more animals on PA14 than on OP50 to compensate for the reduced fertility. We found this insufficient to reliably produce enough embryos for multigenerational experiments, and therefore also delayed the start of training until most animals were young adults (56–60 hr). Importantly, training either late L4 animals (51–52 hr) or young adult animals (57–58 hr) both produced strong PA14 aversion responses in the trained populations (*Figure 1*, *Figure 1—figure supplement 1*). Furthermore, a few *daf-7p::gfp* expression experiments that started with training P0 animals 48–54 hr post bleach were successfully completed. The results of these experiments did not

**Table 1.** Experimental conditions and results for *daf-7p::gfp* expression levels in ASI neurons of wild-type animals.

| Reporter* | Pre-growth temperature | P0-F2 growth temperature | PA14 isolate[†] | Numerical index | F1 elevated *daf-7p::gfp* | F2 elevated *daf-7p::gfp* |
|---|---|---|---|---|---|---|
| MC | 15 °C | 20 °C | B | 1, 2, 3, 4, 5 | Y | N |
| MC | 20 °C | 20 °C | B | 1, 2, 3 | ϒ | N |
| MC | 20 °C | 20 °C | H | 1 [‡] | ϒ | N |
| MC | 20 °C | 20 °C | M | 1, 2, 3 | ϒ | N |
| SC | 20 °C | 20 °C | B | 1 | Y | N |
| SC | 20 °C | 20 °C | H | 1 | Y | N |
| MC | 20 °C | 25 °C | B | 1, 3 | Y | Y |
| MC | 20 °C | 25 °C | B | 2 | Y | N |
| SC | 20 °C | 25 °C | B | 1, 3 | Y | Y |
| SC | 20 °C | 25 °C | B | 2, 4 | Y | N |

*MC, multi-copy reporter in strain FK181; SC, single-copy reporter in strain QL296.

[†]B, Balskus lab isolate; H, Hunter lab isolate; M, Murphy lab isolate.

[‡]Prepared training plates were stored at 4°C for 48 hr prior to start of training.

**Table 2.** Experimental conditions and results for PA14 avoidance (Choice) assay.

| Experiment | Genotype | Growth temp | PA14 isolate* | Light (L) Dark (D) | Azide (A) Cold (C) | F1 PA14 avoidance | F2 PA14 avoidance | Notes |
|---|---|---|---|---|---|---|---|---|
| 1.1 | N2 | 20 °C | B | L | C | Y | N | † |
| 1.2 | N2 | 20 °C | B | D | C | NA | N | † |
| 2 | N2 | 20 °C | B | L | A | N | N | ‡ |
| 3 | N2 | 20 °C | M | L | A | Y | N | |
| 4.1 | N2 | 20 °C | B | L | A | N | N | § |
| 4.2 | N2 | 20 °C | B | L | A | N | N | § |
| 5.1 | N2 | 25 °C | B | D | C | Y | N | ¶ |
| 5.2 | N2 | 25 °C | B | D | C | N | Y | ¶ |
| 6 | N2 | 20 °C | B | D | C | N | N | |
| 7 | sid-1(qt9) | 20 °C | B | D | C | N | NA | |
| 8 | sid-1(qt9) | 25 °C | B | L | C | N | NA | |
| 9 | sid-2(qt42) | 20 °C | B | D | C | N | NA | |
| 10 | sid-2(qt42) | 25 °C | B | L | C | N | NA | |

*B, Balskus lab isolate; M, Murphy lab isolate.

†A single biological replicate was split and separately assayed in the light and dark.

‡For the P0 choice assay, half the choice plates were treated with azide and half with cold induced rigor.

§A single biological replicate was split and adult P0 worms and their embryos in one sample were centrifuged prior to bleach treatment with vigorous mixing (Exp 4.2).

¶Two independent biological replicates were trained and assayed in parallel.

differ from those obtained when training began with older animals (*Figure 2*, *Table 1*, *Figure 2—source data 1*). Training young adults greatly improved F1 egg recovery, allowing for more reliable testing of F1 and F2 progeny. Importantly, at the end of the 24 hr training period (80–84 hr since embryo isolation, 20 °C) the naive and trained worm populations had laid many eggs, ensuring that the recovered in utero F1 eggs were fertilized after exposure to PA14. This change in training time usually produced sufficient F1 and F2 animals for measurement of *daf-7p::gfp* expression levels in the ASI neurons and learned avoidance assays. However, the trained and control F2 progeny remained indistinguishable (*Figures 1 and 2*, *Figure 3*, *Tables 1 and 2*).

To control for possible laboratory environmental differences, we also tested variations to the protocol, including worm growth temperature (15 °C, 20 °C, 25 °C) before training (husbandry), during training, and in the F1-F2 generations (*Figures 2 and 3*, *Figure 2—source data 1*, *Figure 3—source data 1*). These growth temperature variations were inspired by variability in reported husbandry conditions (*Moore et al., 2019*; *Kaletsky et al., 2020*; *Moore et al., 2021b*), the report that increased cultivation temperature (25 °C) is known to increase *P. aeruginosa* pathogenicity (*Tan et al., 1999*), and the fact that the experiments that identified the *daf-7p::gfp* response to *P. aeruginosa* were performed at 25 °C (*Meisel et al., 2014*). In line with this, we observed more robust *daf-7p::gfp* expression in ASI neurons in P0 animals exposed to PA14 at 25 °C (*Figure 2*). However, none of these adjustments resulted in robust elevated *daf-7p::gfp* expression in F2 progeny (*Figure 2*, *Table 1*, and *Figure 2—source data 1*). While 25 °C growth enhanced PA14-induced *daf-7p::gfp* expression in P0 animals, the effect on the F2 generation was an apparent increase in variability, as the F2 progeny of PA14 and OP50 trained animals were more likely to show statistically significant differences in *daf-7p::gfp* expression, but in either direction (*Figure 2*). Similarly, modified training and growth conditions to enable inheritance of learned avoidance behavior did not result in significant changes to P0 and F1 inheritance results and did not result in robust detection of learned avoidance in the F2 progeny (*Figure 3*, *Figure 3—source data 1*).

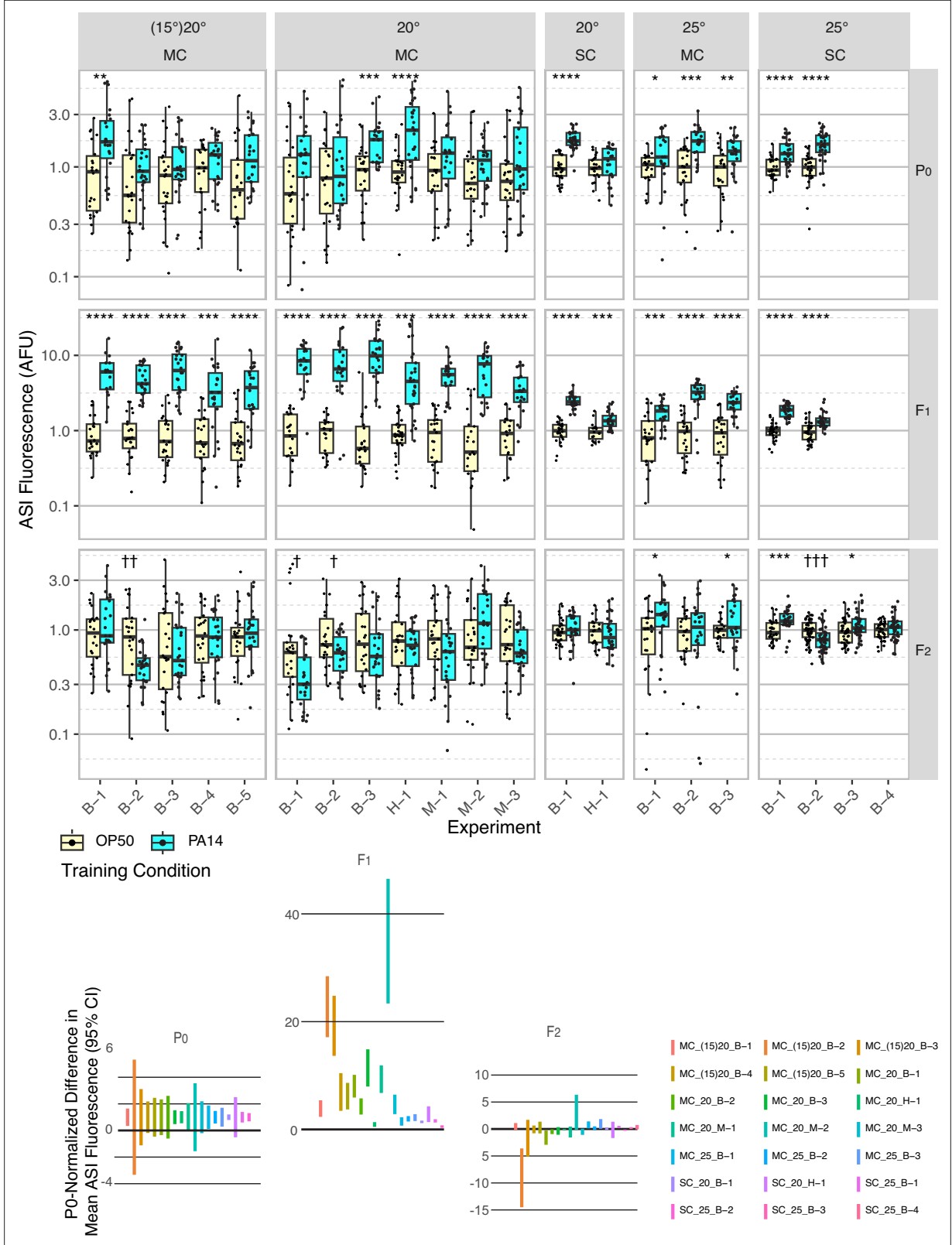

**Figure 2.** Effect of experimental design changes on multigenerational ASI *daf-7p::gfp* expression levels after P0 PA14 exposure. Box plots of results for 21 independent experiments are normalized to the average OP50 value by generation within each experiment for summary presentation (see *Table 1* and *Figure 2—source data 1* for experimental conditions). This figure includes the experiments shown in *Figure 1*. FK181 contains an integrated multi-copy (MC) tandem array composed of the *daf-7p::gfp* reporter and the co-injection marker *rol-6(su1006)* (*Murakami et al., 2001*). QL296 is a

*Figure 2 continued on next page*

*Figure 2 continued*

single copy (SC) insert of *daf-7p::gfp* with *unc-119(+)* as the co-selection marker (**Zhan et al., 2015**). Worms were cultured at either 20 °C or 25 °C and exposed to one of three different PA14 isolates (B, Balskus; H, Hunter; M, Murphy labs).In some experiments, worms were grown at 15 °C for at least three generations prior to the P0 generation (indicated with parentheses, i.e. (15)20). The 95% confidence interval for the predicted absolute difference in means between conditions, normalized to the predicted P0 difference for each experiment (when applicable), is presented in the lower portion of the figure. **Figure 2—figure supplement 1** shows the same data for individual neurons, rather than the mean of the two neurons. Statistical significance **** $p<0.0001$, ***$<0.001$, **$<0.01$, *$<0.05$. Non-significant labels ($p>0.05$) are omitted for clarity. † Indicate statistical significance with control higher than experiment. See Methods section for statistical methods. Data for this figure is presented in **Figure 2—source data 1**.

The online version of this article includes the following source data and figure supplement(s) for figure 2:

**Source data 1.** This file contains *daf-7p::gfp* expression in ASI neurons data plotted in **Figures 1 and 2**, **Figure 2—figure supplement 1**, and **Figure 5**.

**Figure supplement 1.** Box plot display of individual ASI *daf-7p::gfp* expression levels after P0 PA14 exposure.

**Figure supplement 2.** Coefficient of variation analysis of ASI *daf-7p::gfp* expression levels.

## Using a single-copy *daf-7p::gfp* reporter strain did not result in elevated *daf-7p::gfp* expression in ASI neurons among F2 progeny

The strain FK181 contains an integrated multi-copy *daf-7p::gfp* reporter and the co-injection marker *rol-6(su1006)* [pRF4] tandem array (**Murakami et al., 2001**). We noticed sporadic loss of both the Rol-6 phenotype and *gfp* expression from the line, suggesting either spontaneous transgene silencing or changes in the *daf-7p::gfp* reporter copy number. A second, independent stock of FK181 acquired from the CGC displayed similar instability. To characterize this in greater detail, we established and maintained nine independent FK181 lines and observed co-incident loss of the Roller phenotype and *gfp* expression in all nine lines, and in six of nine lines this loss of array phenotypes was heritable (**Table 3**). It is known that growth conditions and mutations that modulate small RNA pathways can affect transgene expression levels from multi-copy tandem arrays (**MacMorris et al., 1994**; **Cui et al., 2006**; **Fischer et al., 2013**). Because this key reagent is not reliable, we obtained QL296, a strain containing an integrated single-copy *daf-7p::gfp* reporter that does not include a co-injection marker with a morphological or growth phenotype (**Zhan et al., 2015**). The QL296 fluorescent readout was less bright than FK181 but reproduced the elevated *daf-7p::gfp* fluorescence in P0 and F1 progeny. Encouragingly, the coefficient of variation was reduced to about 0.27 (**Figure 2—figure supplement 2**), which should increase the sensitivity of this assay to detect differences in *daf-7p::gfp* expression. However, measured *daf-7p::gfp* levels in F2 descendants of trained and control P0 animals carrying the single-copy reporter were usually indistinguishable, as we had seen with the multi-copy reporter (**Figure 2**).

## Performing the learned avoidance assay in light or dark conditions did not produce heritable F2 avoidance

A recent report indicates that visible light contributes to *C. elegans* ability to detect and avoid PA14 (**Ghosh et al., 2021**). We found that choice assays performed in the light (on benchtop) or in the dark (in a closed cabinet) both readily produced learned (P0) and inherited (F1) PA14 avoidance but that F2 inheritance of learned avoidance was not reliably detected in either assay condition (**Figure 3** and **Table 2**). The choice index ([number of animals choosing OP50 - number of animals choosing PA14] / Total number of choices) for assays performed in the light for both trained and control animals were frequently higher than the choice index for assays performed in the dark, however the learning indexes (the relative differences) were indistinguishable. Thus, ambient lighting conditions can impact the measured choice index between control and PA14 trained animals, but they do not detectably disrupt the learning index. Overall, none of the environmental adjustments were sufficient to reliably reproduce the reported results; the summary analysis of the learning indexes for all experiments showed highly significant P0 training results, modest F1 intergenerational inheritance, and insignificant F2 transgenerational inheritance (**Figure 3J**).

## OP50 growth conditions strongly affect OP50 aversion

Naïve (OP50 grown) worms often show a bias towards PA14 in choice assays (**Zhang et al., 2005**; **Ha et al., 2010**; **Moore et al., 2019**; **Pereira et al., 2020**; **Lalsiamthara and Aballay, 2022**). This

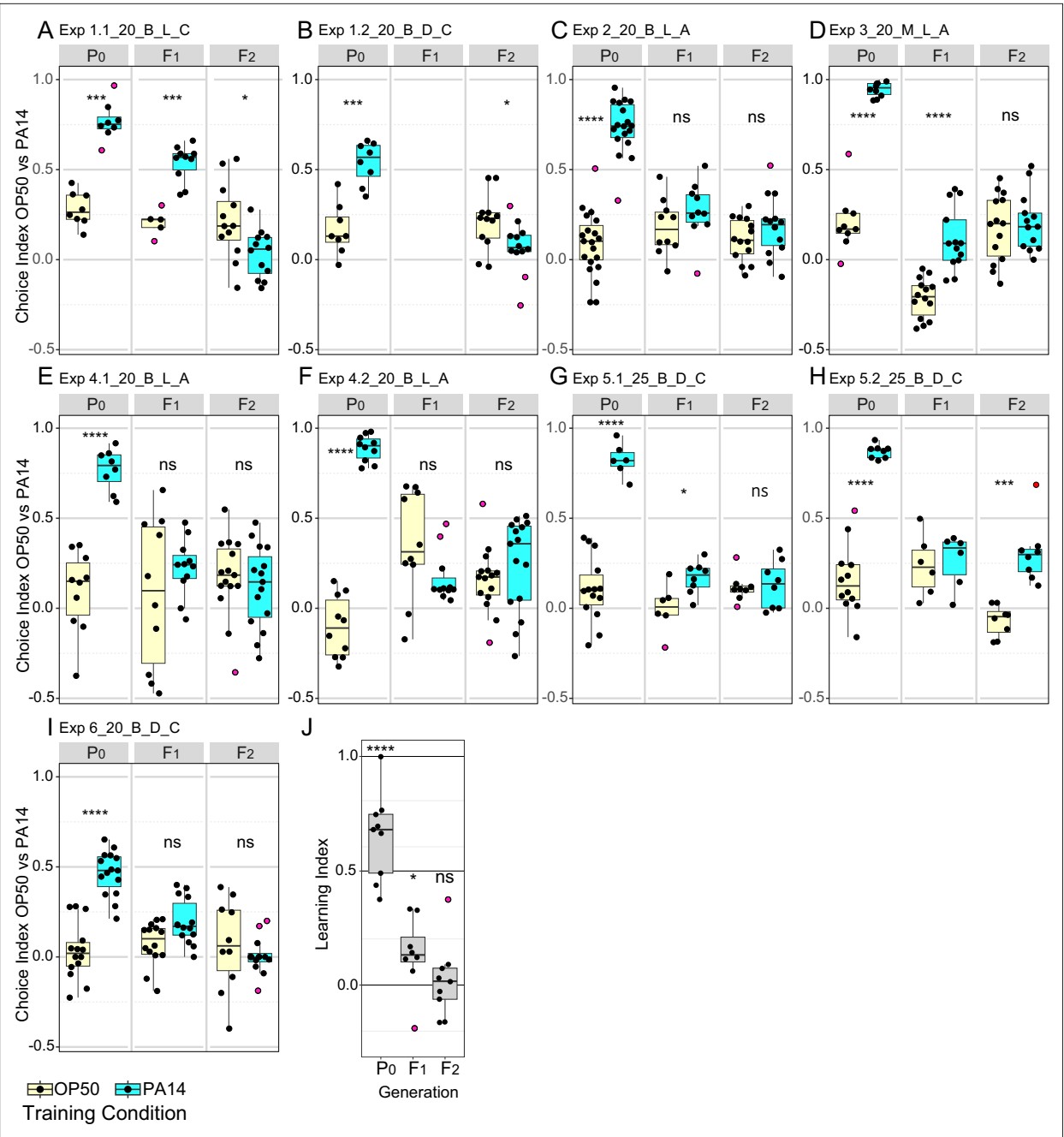

**Figure 3.** Effects of experimental design changes on multigenerational PA14 avoidance after PA14 exposure. Growth and assay conditions for each experiment are described in *Table 2* and *Figure 3—source data 1*. This figure includes the experiments shown in *Figure 1*. Choice index calculated as described in *Figure 1*. (A-I) The experimental and assay conditions are indicated in the panel titles; [Exp #] [growth temperature (20 °C or 25 °C)] [PA14 isolate (B, Balskus or M, Murphy)] [light/dark assay condition (L or D)] azide/cold paralytic (A or C). (J) Summary panel showing learning indexes (choice index for trained minus choice index for control) for all nine results (eight for F1 animals). See *Figure 3—figure supplement 1* for Fisher's exact test analysis of choice assay results. Purple dots in choice assay results indicate outlier data points that were included in all statistical tests. Statistical significance **** p<0.0001, ***<0.001,**<0.01, *<0.05, ns >0.05. See Methods section for statistical methods. Data for this figure is presented in *Figure 3—source data 1*.

The online version of this article includes the following source data and figure supplement(s) for figure 3:

**Source data 1.** This file contains the choice assay data plotted in *Figures 1 and 3*, *Figure 1—figure supplement 2*, *Figure 3—figure supplement 1*, and *Figure 5—figure supplement 2*.

**Figure supplement 1.** P0 PA14 exposure fails to reproducibly induce F2 PA14 avoidance behavior.

*Figure 3 continued on next page*

*Figure 3 continued*

**Figure supplement 2.** Sample size does not correlate with choice index scores.

**Figure supplement 2—source data 1.** This file presents data on the correlation coefficient between number of worms per choice plate and choice index across all experiments shown in *Figure 3*.

response, rather than representing an innate attraction to PA14, likely reflects the context of the worm's recent growth on OP50, a mild *C. elegans* pathogen (*Garigan et al., 2002*; *Garsin et al., 2003*; *Shi et al., 2006*). Thus, the naïve worms presented with a choice between a recently experienced mild pathogen (OP50) and a novel food choice (PA14) initially choose the novel food instead of the known mild pathogen (OP50 aversion). Because the difference in the choice between trained and naïve animals in the P0 generation is highly significant, while the difference in the F1 generation is much reduced (*Figure 3*), even a slight reduction in naïve aversion to OP50 could affect the ability to detect PA14 avoidance in F1 and F2 animals. Indeed, in our experiments (*Figure 3*), the control worms frequently showed a choice index score higher than that reported by *Moore et al., 2019* and *Kaletsky et al., 2020*. In line with our results, some other groups have also reported higher naïve choice index scores (*Lee and Mylonakis, 2017*). This variability in naive choice may reflect differences in growth conditions of either the OP50 or PA14 bacteria. In addition, we note that among the studies that show naive worm attraction to *Pseudomonas* (OP50 aversion), there are extensive methodological differences from the methods in *Moore et al., 2019*; *Moore et al., 2021b*, including differences in bacterial growth temperature, incubation time, whether the bacteria is diluted or concentrated prior to placement on the choice plates, the concentration of peptone in the choice plates, the length of the choice assay, and the inclusion of sodium azide in the choice assays (*Zhang et al., 2005*; *Ha et al., 2010*; *Moore et al., 2019*; *Pereira et al., 2020*; *Lalsiamthara and Aballay, 2022*). Thus, the cause of the variability across published reports is not clear. Furthermore, because OP50 pathogenicity is enhanced by increased *E. coli* nutritive conditions (*Garsin et al., 2003*; *Shi et al., 2006*), the growth of F1-F4 progeny on High Growth (HG) plates (*Moore et al., 2019*), which contain 8 X more peptone than NG plates and therefore support much higher OP50 growth levels, immediately prior to the F1-F4 choice assays may further contribute to OP50 aversion among the control animals. We note that in our hands, in each aversion assay experiment that produced a significant F1 result, the average F1 control choice index score was lower than that detected in the preceding P0 generation where the control animals were trained on NG plates (*Figure 3*). Thus, changes in growth conditions that enhance OP50 aversion (lower choice index score) could magnify the difference between trained and control animals.

**Table 3.** FK181 genetic instability.

| | | Rol Progeny | | non-Rol Progeny | |
|---|---|---|---|---|---|
| FK181 line | Starve-chunk cycle with first observed non-Rol | GFP+ | GFP- | GFP+ | GFP- |
| 1 | 4* | | | | |
| 2 | 4 | | | | +++ |
| 3 | 7 | + | | + | +++ |
| 4 | 7 | +++ | | | |
| 5 | 7 | +++ | | | |
| 6 | 7 | + | | (+)[†] | +++ |
| 7 | 7 | + | | + | |
| 8 | 8 | 241 progeny | 22% non-Rol, 81% GFP- | | | |
| 9 | 8 | 167 progeny | 26% non-Rol, 46% GFP- | | | |

FK181 was grown to starvation (6 days, 20 °C) on a small NG plate and then chunked to a fresh small NG plate and grown to starvation. The first observed non-Rol hermaphrodite (day 3) was a picked to a fresh plate the Rol and GFP phenotypes of the progeny recorded.

*Sterile.

[†]weak GFP.

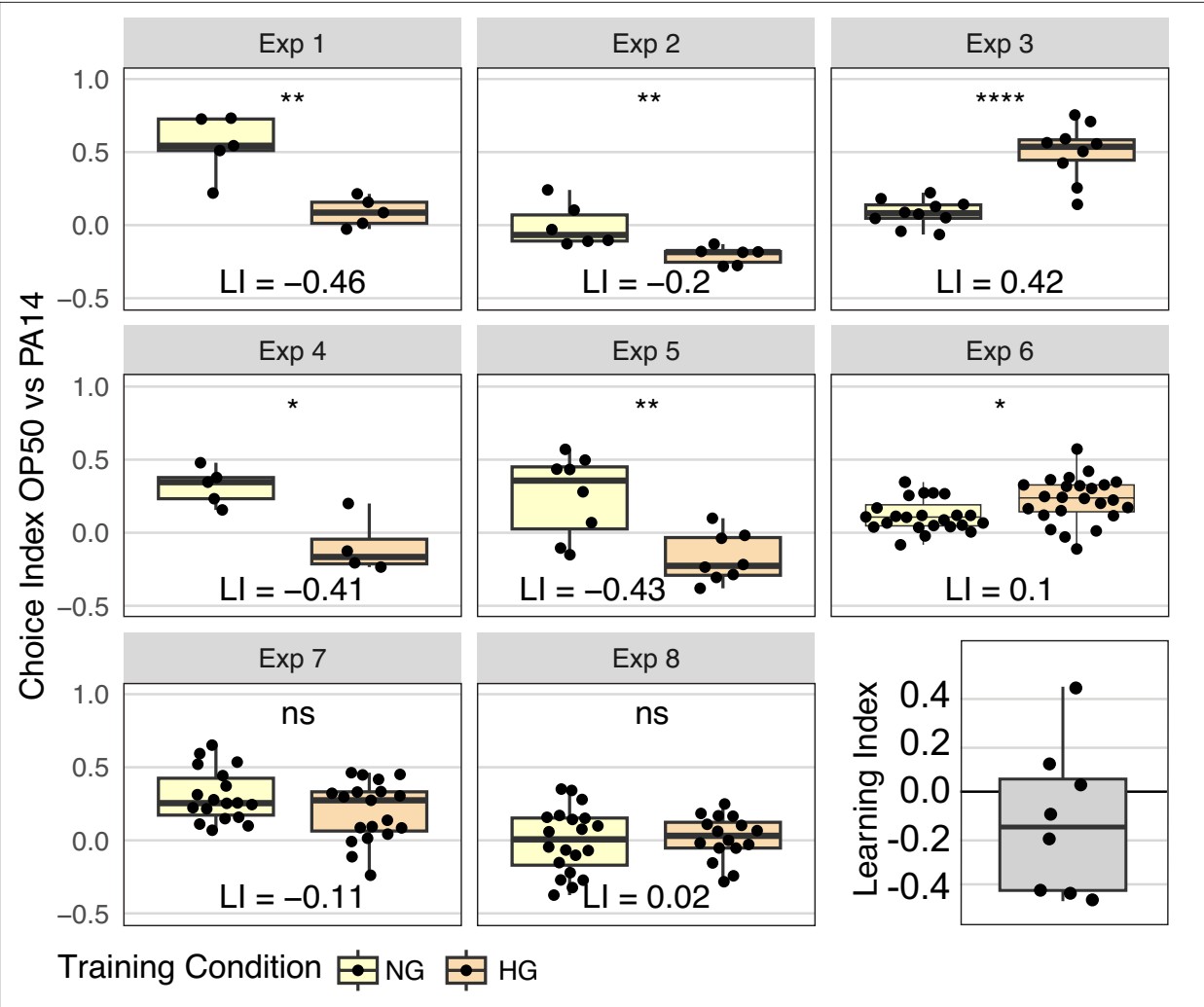

**Figure 4.** Effect of OP50 growth conditions on OP50 aversion. N2 worms grown to adulthood on HG plates were 'trained' on either HG OP50 or NG OP50 plates for 24 hours and then assayed on OP50 vs PA14 choice plates. The choice index was calculated as described in *Figure 1*. The learning index, difference between mean HG choice index and mean NG choice index, for all eight experiments is plotted in the last panel. See *Figure 4— figure supplement 1* for Fisher's exact test analysis of choice assay results. Statistical significance **** p<0.0001, ***<0.001, **<0.01, *<0.05, ns >0.05. See Methods section for statistical methods. Data for this figure is presented in *Figure 4—source data 1*.

The online version of this article includes the following source data and figure supplement(s) for figure 4:

**Source data 1.** This file contains data NG vs HG training condition and choice index on PA14 OP50 choice plates plotted in *Figure 4*, *Figure 4—figure supplement 1*.

**Figure supplement 1.** OP50 growth conditions, independent of PA14 exposure, can affect OP50 vs PA14 choice.

To test the effect of OP50 growth conditions on OP50 aversion, we plated young adult N2 animals from HG OP50 plates on either HG OP50 or NG OP50 plates prepared exactly as for control training plates. After 24 hours the HG OP50 'trained' and NG OP50 'control' animals were tested on standard OP50 vs PA14 choice assay plates. In four experiments, the magnitude of the differences in mean choice index (Learning Index, LI) exceeded 0.4 (*Figure 4*). However, the inter-experiment variability was also high, with two experiments failing to detect a difference and two experiments showing an inverse result to the other four. We note that when analyzing the sum of all worm choices across all choice assay plates by Fisher's exact test (*Figure 4—figure supplement 1*), five of eight experiments show that HG OP50 growth conditions induce OP50 avoidance, and seven of eight experiments show significant differences between worms cultured for 24 hr on NG OP50 and HG OP50 plates. Although the results in most experiments are consistent with the hypothesis that HG OP50 exposure enhances OP50 aversion (negative learning index), due to the variability between experiments we interpret the

results as inconclusive. Even so, these results highlight the sensitivity of the choice assay to environmental differences and demonstrate the not inconsequential difference between NG OP50 and HG OP50 conditions. Although the experimental design implicitly controls for the difference between P0 and F1 generation growth conditions, the magnitude of the NG OP50 vs HG OP50 response, which can exceed the magnitude of the OP50 vs PA14 F1 response but in the opposite direction, suggests that this experimental variable may contribute to the irreproducibility of the published results. Indeed, any enhanced OP50 aversion that results from growing worms on HG OP50 plates likely increases the ability to detect lingering PA14 aversion.

While these observations do not explain our inability to replicate the published results, they do illustrate the sensitivity of the choice assay to differences in bacterial growth conditions. This supports the conjecture that non-obvious growth condition differences, beyond what we have explicitly tested, may contribute to the discrepancy between our results and the previously published results.

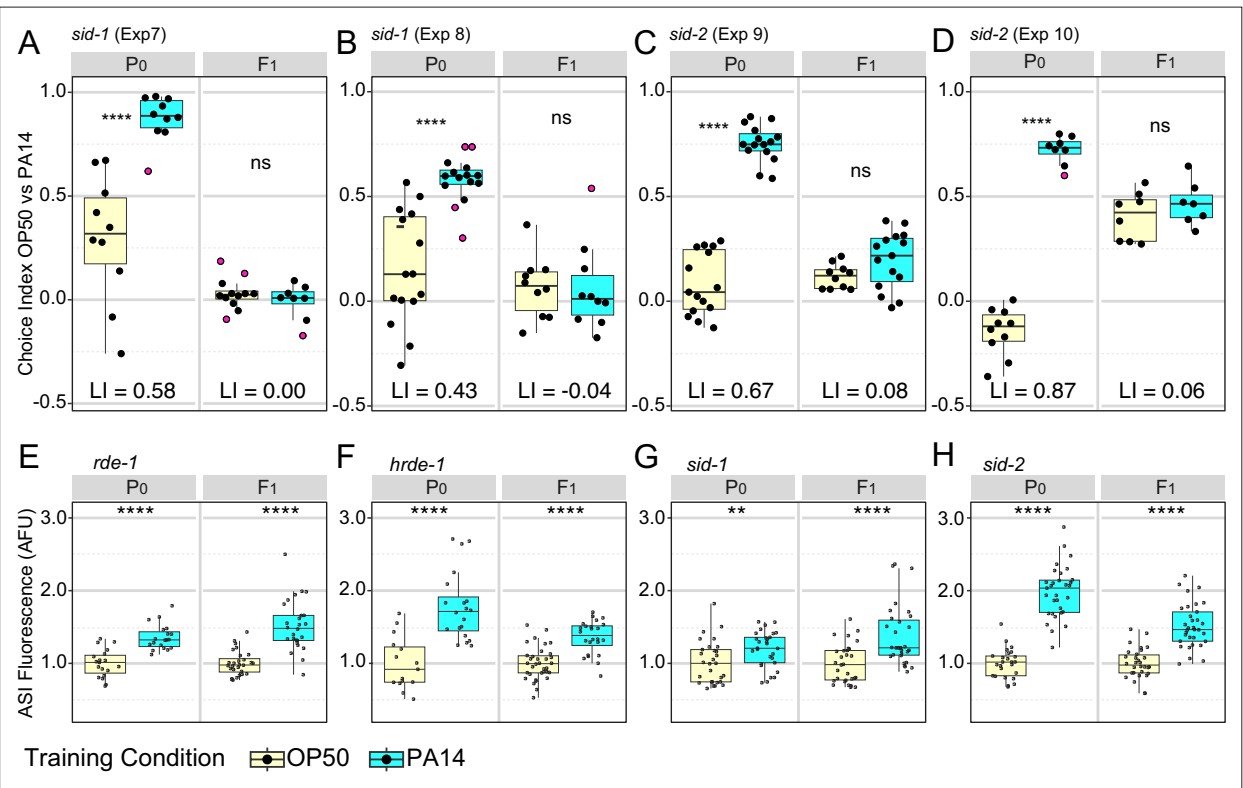

**Figure 5.** Effects of RNAi pathway mutants on intergenerational (**F1**) inheritance of avoidance behavior and ASI *daf-7p::gfp* expression levels. (**A–D**) *sid-1(qt9)* and *sid-2(qt42)* choice assays. Growth and assay conditions for each experiment are described in *Table 2*. The choice index was calculated as described in *Figure 1*. See *Figure 5—figure supplement 1* for Fisher's exact test analysis of *sid-1* and *sid-2* choice assay results. N2 choice assay results performed in parallel with the *sid-1* and *sid-2* experiments are presented in *Figure 5—figure supplement 2*. (**E–H**) ASI expression experiments showing average ASI neuron *daf-7p::gfp* expression levels per worm for each of the four genotypes are shown. ASI *daf-7p::gfp* expression levels are normalized to the average OP50 value by generation within each experiment for ease of presentation. The experiments shown in panels **E-H** were performed with animals cultured and trained at 20 °C. Statistical significance **** p<0.0001, ***<0.001, **<0.01, *<0.05, ns >0.05. See Methods section for statistical methods. Data for this figure is presented in *Figure 2—source data 1*, *Figure 5—source data 1*.

The online version of this article includes the following source data and figure supplement(s) for figure 5:

**Source data 1.** This file contains the data plotted in *Figure 5*, *Figure 5—figure supplement 1*, and *Figure 5—figure supplement 2*.

**Figure supplement 1.** *sid-1* and *sid-2* are required for intergenerational (F1) inheritance of avoidance behavior.

**Figure supplement 2.** N2 P0 and F1 choice assay results performed in parallel to *sid-1* and *sid-2* experiments.

## The systemic RNAi pathway genes *sid-1* and *sid-2* act in parallel or downstream of the neuronal TGF-β pathway for intergenerational (F1) inheritance of pathogen avoidance

We were able to replicate the requirement, as reported (*Kaletsky et al., 2020*), for *sid-1* and *sid-2* in intergenerational (F1) inheritance of learned avoidance. *Kaletsky et al., 2020* used a weak *sid-1(pk3321)* allele that remains fully sensitive to feeding RNAi targeting intestinal *act-5* (*Whangbo et al., 2017*), thus we repeated these experiments with either an early nonsense *sid-1(qt9)* allele (aversion assay) or a cas9 generated deletion of the entire coding region, *sid-1(qt158)* (*daf-7p::gfp* expression assay), both of which eliminate systemic RNAi silencing. We found that *sid-1(qt9)* P0 animals learn to avoid PA14 (learning index 0.58 and 0.43) but that their F1 progeny showed no inherited learned avoidance (learning index –0.04 and 0.00; *Figure 5A and B*). Similarly, we found that *sid-2(qt42)* P0 animals also learn to avoid PA14 (learning index 0.87 and 0.67) while their F1 progeny showed little inherited learned avoidance (learning index 0.06, 0.08; *Figure 5C and D*). Unexpectedly, the exogenous RNAi (*rde-1*), heritable RNAi (*hrde-1*), systemic RNAi (*sid-1*), and feeding RNAi (*sid-2*) pathways were not required for intergenerational inheritance of elevated *daf-7p::gfp* expression in the F1 progeny of PA14-trained animals (*Figure 5E–H*). Since Moore et al., (2019) showed that like *sid-1* and *sid-2* mutants, *daf-7* mutant P0 animals fail to transmit learned avoidance to their F1 progeny, we conclude that *sid-1* and *sid-2* must act in parallel or downstream of the neuronal TGF-β pathway for F1 inheritance of learned avoidance, not upstream as proposed by *Kaletsky et al., 2020*. If learned avoidance and the neuronal TGF-β pathway act in parallel, then the relative strength and time of activation of the two responses may be separately regulated. This is in line with our observation that identical training conditions (20 °C) produce more reliable aversion behavior than *daf-7p::gfp* upregulation in the P0 generation yet more reliable *daf-7p::gfp* upregulation than aversion behavior in the F1 generation.

## Reanalysis of small RNA seq data identifies candidate siRNA-regulated pathogen response genes

The RNAi mutant analysis (*Figure 5*) confirmed the results by *Moore et al., 2019* and *Kaletsky et al., 2020* that *sid-1* and *sid-2*, which are required for systemic and feeding RNAi, are important for intergenerational inheritance of PA14 avoidance. Since antisense endo-siRNAs have been implicated in heritable gene silencing (*Minkina and Hunter, 2018*; *Duempelmann et al., 2020*), PA14 induced changes in small RNAs targeting mRNAs that respond to PA14 exposure may identify heritable RNAi-regulated pathogen response pathways. To investigate the contribution of regulatory small RNAs to heritable pathogen avoidance, *Moore et al., 2019* sequenced mRNAs and small RNAs from control and PA14 trained P0 animals. The assembled small RNA sequencing libraries contain both sense-strand (piRNAs or 21U-RNAs and miRNAs) and antisense-strand (endogenous siRNAs) small RNAs, yet the published work appears to have presented only findings on the sense-strand small RNAs. To determine whether PA14-induced changes in antisense small RNA levels may contribute to heritable pathogen responses, we reanalyzed the sequencing data from *Moore et al., 2019*. We detected sense-strand small RNAs that correspond to 2041 genes that increased (1429) or decreased (612) by twofold or more (Padj. ≤0.05; *Figure 6A*, *Figure 6—figure supplement 1A*), which, using our pipeline, is similar to the mapping data published by *Moore et al., 2019* (1252 up and 450 down). We also mapped differentially expressed antisense small RNAs to over 4000 genes (3928 up ≥ twofold and 171 down > twofold [Padj. ≤0.05]; *Figure 6B*, *Figure 6—figure supplement 1B*). We then plotted the predicted mRNA targets that changed by twofold or more (P adj. ≤0.05) against the antisense small RNAs that changed by twofold or more (P adj. ≤0.05), identifying 116 mRNAs that are putatively regulated by the endogenous RNAi pathway in response to PA14 exposure (*Figure 6C*). Curiously, the *maco-1* gene (red dot in *Figure 6C*), identified as the regulatory target for the piRNA mediated multi-generational response to PA14 exposure (*Kaletsky et al., 2020*), is not targeted by many differentially expressed antisense endo-siRNAs. Thus, the effect of the *P. aeruginosa*-expressed small RNA P11 on *maco-1* expression, which was identified in a subsequent study (*Kaletsky et al., 2020*), is not likely to be directly mediated by the endogenous RNAi pathway.

We then analyzed gene ontology descriptions of the detectably expressed genes (*Schindelman et al., 2011*) to determine whether any of the subset of genes potentially regulated by the endo-RNAi pathways are likely to contribute to heritable pathogen aversion. We found that 257 of the 14,194

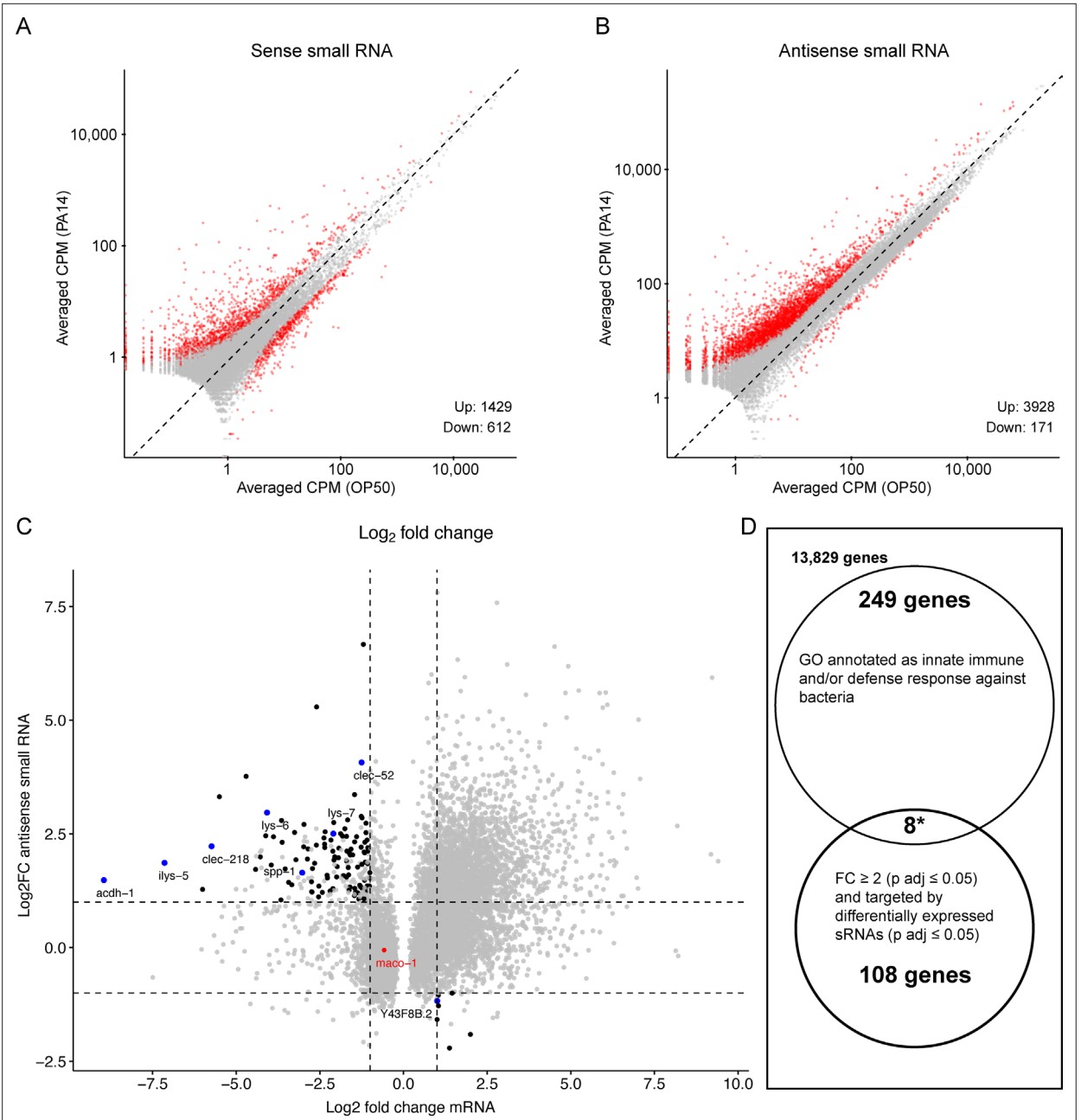

**Figure 6.** Re-analysis of small RNA and mRNA sequence data from PA14 exposed and control P0 animals. RNA sequence data (PRJNA509938) was downloaded and re-analyzed as described in Methods. (**A, B**) Scatter plots comparing sense and antisense strand small RNAs (17–29 nucleotides) from worms grown on PA14 and OP50. Red dots represent ≥twofold change and Benjamini-Hochberg corrected p values (P adj. ≤0.05) and grey dots correspond to less than a twofold change or insignificant difference (P adj. >0.05). Volcano plots of the same data are shown in *Figure 6—figure supplement 1*. (**C**) Black dots represent genes associated with small RNAs that show a significant twofold or greater change in abundance (Y axis) and map to an mRNA that also shows a significant twofold or greater change in abundance (X axis). Blue dots correspond to the subset of target mRNAs associated by GO analysis (panel D) with anti-bacterial or innate immune responses. The red dot represents the *maco-1* gene. (**D**) Gene Ontology analysis of differentially expressed mRNAs targeted by differentially expressed small RNAs. Eight of the 116 small RNA targeted differentially expressed mRNAs are also associated with antibacterial or innate immune responses, while 257 of all 14,194 detectably expressed genes share similar GO annotations representing a 3.8-fold (p<0.0001, hypergeometric distribution) enrichment over neutral expectations.

The online version of this article includes the following source data and figure supplement(s) for figure 6:

**Source data 1.** This file contains re-analyzed differential gene expression results of RNA-seq data from PRJNA509938 (*Moore et al., 2019*).

**Figure supplement 1.** Volcano plot displays reanalyzed small RNA sequence data from PA14 exposed and control P0 animals.

**Table 4.** Putative endo-siRNA targeted genes implicated in host-pathogen responses.

| Gene | Summary description of predicted and/or documented activities |
| --- | --- |
| Y43F8B.2 | Predicted to enable kinase regulator activity and protein kinase A binding activity. Involved in innate immune response. |
| ilys-5 | Predicted to enable lysozyme activity. Predicted to be involved in defense response to Gram-positive bacterium. |
| acdh-1 | Predicted to enable acyl-CoA dehydrogenase activity. Involved in defense response to Gram-negative bacterium and innate immune response. |
| clec-218 | Involved in defense response to Gram-positive bacterium. |
| lys-6 | Predicted to enable lysozyme activity. Predicted to be involved in innate immune response and signal transduction. |
| lys-7 | Involved in defense response to other organisms. |
| clec-52 | Predicted to enable signaling receptor activity. Involved in defense response to Gram-positive bacterium. |
| spp-1 | Enables pore-forming activity. Involved in defense response to other organism and pore formation in membrane of another organism. Part of pore complex. |

Summarized descriptions from WormBase (WS291).

detected expressed genes are annotated as either innate immune response or defense response against bacteria (*Figure 6D*). Eight of the 116 genes putatively regulated by endo-siRNAi pathways (blue dots in *Figure 6C*; 3.8-fold enrichment) are so annotated (*Table 4*). This analysis demonstrates that small RNA regulated genes are differentially expressed in response to pathogen exposure. While the identified genes are candidates for mediating a heritable pathogen aversion response, to support this conjecture it will be necessary to determine whether they and their candidate small RNA regulators are differentially regulated in the F1 and F2 progeny of pathogen exposed animals.

## Summary of thoughts and concerns regarding the potential causes of the irreproducibility

Likely causes for the discrepancy between our results and the published reports include uncontrolled environmental variables or genetic drift (microbes or worms). To control for this, we included in our investigation independent isolates of OP50 and PA14, we cultured worms at different growth temperatures, and we prepared bacterial samples under different growth regimes (aeration vs standing liquid cultures). We note that previous bacterial RNA sequence analysis identified a small non-coding RNA called P11 whose expression correlates with bacterial growth conditions that induce heritable avoidance (*Kaletsky et al., 2020*). Critically, *C. elegans* trained on a PA14 ΔP11 strain (which lacks this small RNA) still learn to avoid PA14, but their F1 and F2-F4 progeny fail to show an intergenerational or transgenerational response (Figure 3L in *Kaletsky et al., 2020*). The fact that we observed an intergenerational (F1) avoidance response is evidence that our PA14 growth conditions induce P11 expression. We also confirmed that our PA14 growth conditions induced strong pathogenicity (*Figure 1—figure supplement 2*). Furthermore, we showed that OP50 control culture conditions can dramatically affect naïve worm behavior in the PA14-OP50 choice assay (*Figure 4*), supporting the conjecture that environmental factors may contribute to the observed discrepancy. Some environmental differences may be systemic and rooted to laboratory or geographical constraints, including humidity, which could affect salinity levels, lysogen activation, or the presence of other, potentially contaminating bacteria in the environment reflecting adjacent laboratory activities. If these potential environmental differences are sufficient to obscure the detection of the transgenerational effect, then the robustness and ecological significance of the transgenerational effect in a natural setting must be minor.

The imaging-based assay measures the expression of a multi-copy *daf-7p::gfp* transgene. We noted that although this transgene array is integrated, the FK181 strain must be monitored for maintenance of the Rol phenotype, which correlates with detection of *gfp*. The instability of the FK181 strain may reflect sporadic transgene silencing or transgene-array rearrangements resulting in lower transgene expression. These observations illustrate that this key reagent is not reliable. To control for this, we obtained the published single-copy non-Rol *daf-7p::gfp* strain QL296 (*Zhan et al., 2015*). Using this single-copy *daf-7p::gfp* reporter we detected a robust difference in the P0 and F1 generations, but did not detect significant differences in the F2 generation (*Figure 2*, *Figure 2—source data 1*).

Ideally, we would have compared our results with the published results (*Moore et al., 2019*), to possibly identify additional experimental parameters for further investigation; for example, a quantitative comparison of naïve choice in the P0 and F1 generations could help to determine the role of bacterial growth in the choice assay response. However, none of the raw data for the published figures and unpublished replicate experiments (*Moore et al., 2019*) were available on the publisher's website or provided upon request to the corresponding author. In the absence of a quantitative comparison, it remains possible that an explanation for the discrepancies between our results and those of *Moore et al., 2019* has been overlooked.

### Concluding remarks

Although we cannot explain the cause of the differences between our results and the published results, we can confidently conclude that this example of TEI is, at present, insufficiently robust for experimental investigation of the mechanisms of multi-generational inheritance. This does not negate the possibility that future investigations may reveal a critical experimental or environmental variable that enables robust multigenerational inheritance. Our attempts to troubleshoot the protocol eliminated several variables, including growth temperature, developmental timing, genetic drift of either WT (N2) or reporter strains (FK181 and QL296) or *P. aeruginosa* strain PA14 (all refreshed from independent stocks). In addition, we determined that some environmental differences associated with the choice assay (light), although potentially altering the choice index, had similar relative effects on trained and control animals (the learning index). PA14 is a lethal pathogen and OP50 is a mild pathogen, thus the most likely cause for the discrepant results is a subtle difference in OP50 physiology in the post P0 generations that affects the magnitude of OP50 aversion. Indeed, we showed that 24 hr exposure to slightly more pathogenic OP50 can dramatically enhance OP50 aversion (*Figure 4*).

While we were unable to reliably replicate the reported F2 aversion or *daf-7p::gfp* expression responses, we confirmed that the F1 aversion response does require the dsRNA transporters *sid-1* and *sid-2*. While this remains a fascinating result, we would not presume that the transported substrate is therefore an RNA molecule, particularly because the systemic RNAi response supported by *sid-1* and *sid-2* is via long double-stranded RNA. To date, no evidence suggests that either protein efficiently transports small RNAs, particularly single-stranded RNAs (*Feinberg and Hunter, 2003*; *Shih and Hunter, 2011*; *McEwan et al., 2012*). How these two proteins support intergenerational epigenetic inheritance remains a mystery. Unfortunately, since neither *sid-1* nor *sid-2* is required for the robust intergenerational increase in *daf-7p::gfp* expression in ASI neurons, the lack of a robust single-animal assay severely hampers mechanistic investigation using this paradigm.

The heritable behavioral and physiological plasticity induced by pathogens is likely to reveal fundamental evolutionarily and ecologically relevant pathways. However, the history of heritable epigenetic research has frequently been hobbled by the difficulty of controlling all environmental factors and the lack of reproducible phenotypic assays. Thus, independent reproducibility is of paramount concern, and we have tried to be completely transparent as a model for how heritability research should be presented within the *C. elegans* community.

## Methods

### Key resources table

| Reagent type (species) or resource | Designation | Source or reference | Identifiers | Additional information |
|---|---|---|---|---|
| Strain, strain background (*Caenorhabditis elegans*) | N2 | *Brenner, 1974* | WT | Obtained from CGC reference 257 |
| Strain, strain background (*Caenorhabditis elegans*) | FK181 | *Murakami et al., 2001* | *ksIs2 [daf-7p::gfp +rol-6(su1006)]* | Obtained from CGC |
| Strain, strain background (*Caenorhabditis elegans*) | QL296 | *Zhan et al., 2015* | *drcSi89 [daf-7p::GFP; unc-119(+)]* | Obtained from Queelim Ch'ng |
| Strain, strain background (*Caenorhabditis elegans*) | HC445 | *Winston et al., 2002* | *sid-1(qt9)* | |
| Strain, strain background (*Caenorhabditis elegans*) | HC306 | *Winston et al., 2007* | *sid-2(qt42)* | |

*Continued on next page*

*Continued*

| Reagent type (species) or resource | Designation | Source or reference | Identifiers | Additional information |
|---|---|---|---|---|
| Strain, strain background (*Caenorhabditis elegans*) | HC1221 | This study | *rde-1(ne219); drcSi89* | |
| Strain, strain background (*Caenorhabditis elegans*) | HC1222 | This study | *hrde-1(tm1200); drcSi89* | |
| Strain, strain background (*Caenorhabditis elegans*) | HC1223 | This study | *sid-1(qt158); drcSi89* | |
| Strain, strain background (*Caenorhabditis elegans*) | HC1218 | This study | *sid-2(qt42); drcSi89* | |
| Strain, strain background (*Escherichia coli*) | OP50 (H) OP50 (M) | *Brenner, 1974* | | OP50 (H) obtained from CGC OP50 (M) obtained from Coleen Murphy |
| Strain, strain background (*Pseudomonas aeruginosa*) | PA14 (H) PA14 (B) PA14 (M) | | | PA14 (H) originated from Ausubel lab. PA14 (B) obtained from Balskus lab. PA14 (M) obtained from Coleen Murphy |
| Chemical compound, drug | Agar | Difco, BD | Cat # 214010 | |
| Chemical compound, drug | Bacto peptone | Gibco, Thermo Fisher Scientific | Cat # 211677 | |
| Chemical compound, drug | 5% Sodium hypochlorite (NaClO) | J.T. Baker, | Cat # 9416–01 | Stored in the dark at 4 °C (less than 3 months old; replace earlier if the resulting embryo prep displays low viability). |
| Software, algorithm | FIJI (1.53i-1.54f) | *Schindelin et al., 2012* | | |
| Software, algorithm | R (4.3.1–4.3.2) | *R Development Core Team, 2023* | | |
| Software, algorithm | RStudio (2023.09.01) | *Posit Team, 2023* | | |
| Software, algorithm | Ggplot2 (3.4.4, 3.5.1) | *Wickham, 2016* | | |
| Software, algorithm | Dplyr (1.1.2, 1.1.4) | *Wickham et al., 2023* | | |
| Software, algorithm | DESeq2 (1.42.1) | *Love et al., 2014* | | |
| Software, algorithm | EnhancedVolcano (1.20.0) | *Blighe, 2023* | | |
| Software, algorithm | Scales (1.3.0) | *Wickham et al., 2023* | | |
| Software, algorithm | Ggpubr (0.6.0) | *Kassambra, 2023* | | |
| Software, algorithm | Grid (4.3.2) | *R Development Core Team, 2023* | | |

The protocol as described in *Moore et al., 2019*; *Kaletsky et al., 2020*; *Moore et al., 2021b*; *Sengupta et al., 2024* was followed closely, with a few clarifications and updates, as detailed in *Supplementary file 1*.

## Bacterial growth and plating

Bacterial cultures were seeded on NG (Normal Growth) or HG (High Growth) plates. NG media: per liter: 3 g NaCl, 17 g agar, and 2.5 g peptone in $H_2O$, autoclave, cool to 55 °C then add 1 mL cholesterol (5 mg/mL), 1 mL 1 M $CaCl_2$, 1 mL 1 M $MgSO_4$, 25 mL 1 M $KPO_4$ (pH 6.0; *Brenner, 1974*). HG media: per liter: 3 g NaCl, 30 g agar, and 20 g peptone in $H_2O$, autoclave, cool to 55 °C then add 1 mL cholesterol (5 mg/mL), 1 mL 1 M $CaCl_2$, 1 mL 1 M $MgSO_4$, 25 mL 1 M $KPO_4$ (pH 6.0; based on *Rose et al., 1982*). NG plates minimally support OP50 growth, resulting in a thin lawn that facilitates visualization of larvae and embryos. HG plates (8 X more peptone) support much higher OP50 growth, resulting in a thick bacterial lawn that supports larger worm populations.

For the choice assay, OP50 for seeding HG growth plates prior to training was grown at room temperature (2 days) without aeration and used fresh or stored at 4 °C for up to 1 month. Seeded HG plates were incubated at RT for 2–4 days before use or incubated for 2–4 days and stored at 4 °C for up to 14 days. Only freshly grown OP50 (37 °C with aeration, 16–22 hours) was used for aversion training and choice plates and was used at growth density or diluted in LB to OD 1.0. For the

*daf-7p::gfp* ASI experiments, we used OP50 grown either at 37 °C with aeration and diluted to 1.0 OD/ml or at room temperature without aeration for growth and training plates (see *Figure 2—source data 1*), again without effect on the results (*Figure 2—source data 1*, *Figure 2*, *Figure 2—figure supplement 1*). PA14 for seeding training plates for both the avoidance assay and *daf-7p::gfp* ASI experiments and for preparation of choice assay plates was grown for 14–19 hr at 37 °C with aeration and diluted to OD 0.5 or 1.0 in LB broth before plating. Whether PA14 cultures were grown for 14 or 18 hr had no discernable effect on pathogenicity (*Figure 1—figure supplement 2*). OP50 and PA14 seeded training plates and choice plates were incubated at 25 °C for 2 days, and then equilibrated to room temperature before use. PA14 and OP50 training plates were incubated either in separate incubators and/or separate partially covered boxes within the same humidity controlled (<50% relative humidity) room during worm training.

## Worm growth and synchronization

Wild-type (N2, FK181, QL296) and mutant worms (Key Resources Table) were grown on NG or HG OP50 plates without starvation, crowding, or contamination for a minimum of three generations before hypochlorite treatment to obtain P0 embryos. Fresh hypochlorite solution was prepared immediately prior to each use. Adult worms were pelleted or allowed to settle (15 mL tube), resuspended in 5–10 mL of hypochlorite solution, and, to minimize contact time with the hypochlorite solution, mixed continuously by nutating or vortexing until less than 5% of the adult body parts were visible. In some experiments, as the adults were initially breaking open (~3–5 min) and the solution began to acquire a yellow tint, the worms and released embryos were pelleted, resuspended in fresh hypochlorite solution, and mixed for an additional 2–3 min, until less than 5% of adult body parts were visible. The released embryos were pelleted and washed 4 X in 5–10 mL of M9 buffer. In the ASI assays, the last M9 wash optionally contained 0.01% Triton X100 to inhibit embryos from sticking to the plastic. We note that the published protocols assert that worms must not be centrifuged immediately prior to bleach treatment and that bleach-treated embryos must be handled gently. Our results do not support this assertion, as the adult worms bleached to prepare P0, F1, and F2 embryos for experiments 4.1 and 4.2 (*Figure 3*, *Figure 3—source data 1*) were treated gently or centrifuged and vortexed during bleaching respectively and showed no meaningful differences between the results.

In all experiments, animals from one or several pooled training plates were treated as a single biological replicate, with a portion of the animals assayed and a portion used for propagation for the next generation. We note that experiments 5.1 and 5.2, which were performed in parallel using a homogenous starting population that was split and then trained and assayed in parallel, showed statistical variation (*Figure 3*, *Table 2*).

## Measuring *daf-7p::gfp* levels in ASI neurons

Both wild-type *daf-7p::gfp* strains (FK181, QL296) and QL296-derived RNAi pathway mutants (HC1218, HC1221, HC1222, and HC1223) were maintained and trained on OP50 or PA14 as described above. Immediately before each respective imaging session approximately 30–40 trained or control animals were transferred by platinum wire directly to 5 µL of 2.5–10 mM levamisole in M9 on 10% agarose pads. A coverslip was added and sealed with beeswax. For the FK181 strain, GFP z-stacks were collected at 1 µm intervals at ×63 magnification on a Zeiss spinning disc confocal microscope. For the QL296 strain, which has dimmer GFP than FK181, GFP z-stacks were collected at 0.5 µm intervals at 63 x magnification on a Zeiss LSM 980 confocal microscope (Harvard Center for Biological Imaging). For data analysis, a maximum intensity projection (MIP) was generated for each z-stack and MIPs across all conditions and generations were then blinded. If the ASI neurons overlapped, then MIPs were generated from subsets of levels in the z-stacks without overlap and scored separately. Each ASI neuron was manually selected from each MIP using the ImageJ polygon selection tool (*Schindelin et al., 2012*). The mean pixel intensity was recorded from each and normalized to nearby background for each MIP. Although we also visually confirmed the expected PA14-induced *daf-7p::gfp* expression in the ASJ neurons of P0 animals, the fluorescence of these neurons was never quantified, as the ASJ response is not induced beyond the P0 generation (*Moore et al., 2019*). In the rare event that an ASI neuron was out of the frame or z-range in the collected z-stacks, the animal was excluded from analysis prior to quantification. We noted bimodality in the distributions of ASI *gfp* expression levels in our data, which likely reflects the position of the bilateral

ASI neurons relative to the objective. To compensate for this, we used the average GFP level per pair of neurons in each worm (*Figures 1 and 2*, and *Figure 5*). We note that measured GFP expression levels in each ASI neuron within an animal are not independent, thus using the average avoids the artifactual bimodal distribution and reports the actual sample size for statistical purposes. We also note that coefficient of variation values using the average ASI level per worm were 20% lower for FK181 and 50% lower for QL296 (*Figure 2—figure supplement 2*) which should increase the ability to detect differences between trained and control animals. Although using the average for each neuron pair or treating each neuron as an independent sample altered the statistical significance of some experiments, it did not engender an F2 response (*Figure 2—figure supplement 1*). All collected data is presented in *Figure 2—source data 1*. Within each analysis pipeline, Z-scores were calculated from normalized intensity values for all samples within a data set, and any data point with a $|Z|>3$ was removed. Unpaired, two-tailed Welch's t-test was performed to generate all reported p-values. Statistics and figures were prepared using R Studio and refined using Adobe Illustrator (*Wickham, 2016*; *Posit Team, 2023*; *R Development Core Team, 2023*; *Wickham et al., 2023*).

## Choice assay conditions, sample size, statistical analysis, and multigenerational propagation

Sodium azide is historically used to preserve transient worm choices in arena chemotaxis assays. In the food choice assay, the effect of the sodium azide can paralyze worms before they enter the bacteria lawn, possibly interfering with their choice. Azide also affects bacteria, potentially affecting the production of molecules that attract or repel worms and thus altering worm choice independent of the paralytic effect on worms. To determine whether sodium azide may influence worm choice, we performed some assays without sodium azide. We found that independent of the application of azide, most worms had made a choice within 30 min and that essentially no worm left their initial food choice during the first hour (data not shown). These no-azide assay plates were moved to 4 °C after 30–60 min to preserve the initial food choice of the worms. After cold-induced rigor was achieved, individual assay plates were returned to room temperature for immediate counting (*Supplementary file 1*). Individual worms were removed as counted to ensure complete and accurate counts. The addition of azide had no discernable effect on the choice assay results (*Figure 1—figure supplement 3*). Which method was used to preserve the worm's initial choice is noted in *Table 2*, *Figure 3—source data 1*, *Figures 1 and 3*.

For comparison to published results, we present the choice assay results in quartile box plots and report a Wilcoxon unpaired P-value for choice assays. We used a one-sample two-tailed T-test to calculate the p-value associated with the summary figure plotting the learning indices for each experiment (*Figure 3J*). The reporting of individual choice assay results as single data points in quartile box plots assumes that there are differences between choice plates that need to be included in significance tests and are independent of the number of worms assayed, limiting the power of the statistical analysis by the number of choice plates. Indeed, statements in *Moore et al., 2019* and *Kaletsky et al., 2020* suggest that some choice assay plates in these studies may have sampled as few as 10–20 worms per assay plate. However, under the assumption that there are no meaningful environmental differences between choice assay plates, the relevant outcome for the experimental design is the overall sum of choices across all choice plates. Here, the null model is that, independent of training condition, each worm chooses OP50 over PA14 with a probability preference p. Testing n worms, independent of the number of animals per choice plate or number of choice plates, the number that choose OP50 is X=pn, with X being binomially distributed. These results can be analyzed by a 2X2 contingency table: two conditions (OP50 vs PA14 training) and two outcomes (OP50 or PA14 choice). In this analysis, Fisher's exact test can be used to test the effect of training on the probability preference p. The results of this alternative analysis of our data are presented in *Figure 3—figure supplement 1*, *Figure 4—figure supplement 1*, and *Figure 5—figure supplement 1*.

The Star Protocol (*Moore et al., 2021b*) cautioned that crowding may influence choice, but we found no difference between choice index scores and sample size (*Figure 3—figure supplement 2*). For each experiment (generation, training condition) we calculated Pearson's correlation coefficient (r) for the number of worms per assay plate against choice index and then plotted r against the average number of worms per plate for each experiment.

R Studio was used for statistical tests and generation of graphical plots which were modified for presentation using Adobe Illustrator (*Wickham, 2016*; *Posit Team, 2023*; *R Development Core Team, 2023*; *Wickham et al., 2023*).

## RNA-seq data analysis

PRJNA509938 RNA-seq data (*Moore et al., 2019*) were re-analyzed from the level of raw sequence reads. The reads were trimmed of universal adaptors using *cutadapt* (*Martin, 2011*) and aligned to the *C. elegans* WBcel235 genome assembly using *bowtie* aligner (*Langmead et al., 2009*) with no more than 2 mismatches allowed (v=2). Aligned reads were counted using *HTSeq-count* (*Anders et al., 2015*; union mode) with settings *stranded = no* for mRNA-seq data and *stranded = yes* (for reads mapped to the same strand as the genomic feature) or *stranded = reverse* (for reads mapped to the opposite strand as the genomic feature) for small RNA data. Differential expression analysis was carried out using DESeq2 R package (*Love et al., 2014*). The absolute value of log2FC ≥1 and Benjamini-Hochberg corrected p-value ≤0.05 were used to define significant changes in gene expression or small RNA levels. Volcano plots were built with EnhancedVolcano (*Blighe, 2023*) and ggpubr R packages (*Kassambra, 2023*).

## Pathogenicity assay

The Balskus and Murphy isolates of the PA14 strain were cultured at 37 °C for 14 or 18 hr. OP50 was similarly cultured for 18 hr. Each culture was diluted in LB to OD 1.0 and 0.75 ml spread to cover the entire agar surface of NG plates which were then incubated at 25 °C for 48 hr before addition of N2 adults. To prepare the N2 adults, HG OP50 grown worms (20 °C) were bleached, and their embryos were placed on HG OP50 plates for 72 hr (20 °C). A single population of adults were washed from the plates with M9, allowed to settle, washed once with M9, resuspended at 4–5 worms/µL and 20 µL spotted onto three technical replicate plates for each condition. The plates were then incubated at 20 °C and scored at 24 hr and at 12 hr intervals thereafter until all the PA14 grown animals were dead. Corpses were counted and removed at each interval. Approximately 15–30% of the PA14 cultured worms were found desiccated on the walls of the plates at the 24 hr time point and were not included in the count. All living worms were transferred to freshly prepared plates at 48 and 96 hr. The fraction of surviving worms was determined by (1 – (cumulative number of dead worms at each scoring interval/sum of the number of living and cumulative dead worms at 120 hr)). All PA14 cultured animals were dead by 120 hr.

## Acknowledgements

We thank members of the Hunter lab for ongoing discussions and specifically Nicole Bush and Alexandra Weisman for detailed comments on the manuscript. We also thank L Ryan Baugh for discussions and comments on the manuscript as well as Richard Losick and Sean Eddy for comments on the manuscript. We thank Emily Balskus for providing an independent sample of PA14. We also thank Coleen Murphy for providing samples of PA14 and OP50 and note that Dr. Murphy and colleagues have posted a response to this study (https://doi.org/10.1101/2025.01.21.634111). Some strains were provided by the *Caenorhabditis* Genetics Center, which is funded by NIH Office of Research Infrastructure Programs (P40 OD010440). We thank the Harvard Center for Biological Imaging (RRID:SCR_018673) for infrastructure and support. We also thank WormBase.

## Additional information

### Funding

| Funder | Grant reference number | Author |
| --- | --- | --- |
| National Institutes of Health | GM089795 | Craig P Hunter |
| John Templeton Foundation | 62579 | Craig P Hunter |

| Funder | Grant reference number | Author |
|---|---|---|

The funders had no role in study design, data collection and interpretation, or the decision to submit the work for publication.

## Author contributions

Daniel Patrick Gainey, Formal analysis, Investigation, Methodology, Writing – review and editing; Andrey V Shubin, Software, Formal analysis; Craig P Hunter, Formal analysis, Supervision, Funding acquisition, Investigation, Visualization, Methodology, Writing – original draft, Project administration, Writing – review and editing

## Author ORCIDs

Daniel Patrick Gainey ⓘ http://orcid.org/0000-0002-0000-7187
Craig P Hunter ⓘ https://orcid.org/0000-0002-1456-5657

Reviewer #1 (Public review): https://doi.org/10.7554/eLife.100254.3.sa1
Reviewer #2 (Public review): https://doi.org/10.7554/eLife.100254.3.sa2
Reviewer #3 (Public review): https://doi.org/10.7554/eLife.100254.3.sa3
Author response https://doi.org/10.7554/eLife.100254.3.sa4

# Additional files

## Supplementary files

Supplementary file 1. PA14 training and choice assay protocol. This is Craig Hunter's distillation and clarification of protocols to train and assess learned and inherited PA14 avoidance in N2 animals through the F2 generation. Green highlights indicate known methodological difference from the Star Protocol (*Moore et al., 2021b*).

MDAR checklist

## Data availability

All data generated and analyzed during this study are included in the manuscript and supporting files; source data files have been provided for all figures.

The following previously published dataset was used:

| Author(s) | Year | Dataset title | Dataset URL | Database and Identifier |
|---|---|---|---|---|
| Moore RS | 2018 | Piwi/PRG-1 Argonaute and TGF-beta Mediate Transgenerational Learned Pathogenic Avoidance | https://www.ncbi.nlm.nih.gov/bioproject/PRJNA509938 | NCBI BioProject, PRJNA509938 |

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
