## [Editor Report · eLife Assessment]

This **important** study reports numerous attempts to replicate reports on transgenerational inheritance of a learned behavior – pathogen avoidance – in *C. elegans*. While the authors observe parental effects that are limited to a single generation (also called intergenerational inheritance), the authors failed to find evidence for transmission over multiple generations, or transgenerational inheritance. The experiments presented are meticulously described, making for **compelling** evidence that in the authors' hands transgenerational inheritance cannot be observed. There remains the possibility that different assay setups explain the failure to reproduce previous observations, although the authors present data suggesting that details of the assay are not that significant. There also remains the possibility that differences in culture conditions or lab environment explain the failure to reproduce previous observations, with updates to the paper having further reduced the probability that this applies here. Even if this were the case, it would imply that the original experimental paradigm was dependent on a very specific context. Given the prominence of the original reports of transgenerational inheritance, the present study is of broad interest to anyone studying genetics, epigenetics, or learned behavior.

[As also pointed out by the authors of this study, the authors of the original reports have provided a response on bioRxiv (DOI: https://doi.org/10.1101/2025.01.21.634111).]

---

## [Referee Report · Reviewer #1 (Public review)]

Summary:

The authors report an inability to reproduce a transgenerational memory of avoidance of the pathogen PA14 in *C. elegans*. Instead, the authors demonstrate intergenerational inheritance for a single F1 generation, in embryos of mothers exposed to OP50 and PA14, where embryos isolated from these mothers by bleaching are capable of remembering to avoid PA14 in a manner that is dependent on systemic RNAi proteins sid-1 and sid-2. This could reflect systemic sRNAs generated by neuronal daf-7 signaling that are transmitted to F1 embryos. The authors note that transgenerational memory of PA14 was reported by the Murphy group at Princeton, but that environmental or strain variation (worms or bacteria) might explain the single generation of inheritance observed at Harvard. The Hunter group tried different bacterial growth conditions and different worm growth temperatures for independent PA14 strains, which they show to be strongly pathogenic. However, the authors could not reproduce a transgenerational effect at Harvard. This paper honestly alters expectations and indicates that the model that avoidance of PA14 is remembered for multiple generations is not robust enough to be replicated in all laboratories.

Overall, this paper that demonstrates that one model for transgenerational inheritance in *C. elegans* is not robust. The author do demonstrate an avoidance memory for F1 embryos that could be a maternal effect, and the authors confirm that this is mediated by a systemic small RNA response. There are several points in the manuscript where a more positive tone might be helpful.

Strengths:

The authors note that the high copy number daf-7::GFP transgene used by the Murphy group displayed variable expression and evidence for somatic silencing or transgene breakdown in the Hunter lab, as confirmed by the Murphy group. The authors nicely use single copy daf-7::GFP to show that neuronal daf-7::GFP is elevated in F1 but not F2 progeny with regards to memory of PA14 avoidance, speaking to an intergenerational phenotype.

The authors nicely confirm that sid-1 and sid-2 are generally required for intergenerational avoidance of F1 embryos of moms exposed to PA14. However, these small RNA proteins did not affect daf-7::GFP elevation in the F1 progeny. This result is unexpected given previous reports that daf-7::GFP is not elevated in F1 progeny of sid mutants.

The authors studied antisense small RNAs that change in Murphy data sets, identifying 116 mRNAs that might be regulated by sRNAs in response to PA14. The authors show that the maco-1 gene, putatively targeted by piRNAs according to the Kaletsky 2020 paper, displays few siRNAs that change in response to PA14. The authors conclude that the P11 ncRNA of PA14, which was proposed to promote interkingdom RNA communication by the Murphy group, may not affect maco-1 expression in *C. elegans*, although they did not formally demonstrate this. The authors define 8 genes based on their analysis of sRNAs and mRNAs that might promote resistance to PA14, but they do not further characterize these genes' role in pathogen avoidance. Others might wish to consider following up on these genes and their possible relationship with P11.

Weaknesses:

This very thorough and interesting manuscript is at times pugnacious.

Please explain more clearly what is High Growth media for *E. coli* in the text and methods, conveying why it was used by the Murphy lab, and if Normal Growth or High Growth is better for intergenerational heritability assays.

Comments on revisions:

The authors have done a reasonable job cordially revising this manuscript, and the authors have addressed most reviewer concerns. It is likely that the P11 gene was in some of the PA14 Pseudomonas strains tested, as one was kindly provided by the Murphy group.

---

## [Referee Report · Reviewer #2 (Public review)]

This paper examines the reproducibility of results reported by the Murphy lab regarding transgenerational inheritance of a learned avoidance behavior in *C. elegans*. It has been well established by multiple labs that worms can learn to avoid the pathogen pseudomonas aeruginosa (PA14) after a single exposure. The Murphy lab has reported that learned avoidance is transmittable to 4 generations and dependent on a small RNA expressed by PA14 that elicits the transgenerational silencing of a gene in *C. elegans*. The Hunter lab now reports that although they can reproduce inheritance of the learned behavior by the first generation (F1), they cannot reproduce inheritance in subsequent generations.

This is an important study that will be useful for the community. Although they fail to identify a "smoking gun", the study examine several possible sources for the discrepancy, and their findings will be useful to others interested in using these assays. The preference assay appears to work in their hands in as much as they are able to detect the learned behavior in the P0 and F1 generations, suggesting that the failure to reproduce the transgenerational effect is not due to trivial mistakes in the protocol. The authors provide a full protocol and highlight key deviations from the Murphy lab protocol. The authors provide good evidence that no single protocol modification was sufficient on its own to explain the divergent results. It remains possible that protocol differences affected the assay cumulatively or that other uncontrolled factors were responsible. Nevertheless, the authors provide good evidence that the trans-generational effect reported by the Murphy lab lacks experimental robustness, calling into question its ecological relevance in the wild.

---

## [Referee Report · Reviewer #3 (Public review)]

Summary:

It has been previously reported in many high-profile papers, that *C. elegans* can learn to avoid pathogens. Moreover, this learned pathogen avoidance can be passed on to future generations - up to the F5 generation in some reports. In this paper, Gainey et al. set out to replicate these findings. They successfully replicated pathogen avoidance in the exposed animals, as well as a strong increase in *daf-7* expression in ASI neurons in F1 animals, as determined by a daf-7::GFP reporter construct. However, they failed to see strong evidence for pathogen avoidance or *daf-7* overexpression in the F2 generation. The failure of replication is the major focus of this work.

Given their failure to replicate these findings, the authors embark on a thorough test of various experimental confounders that may have impacted their results. They also re-analyze the small RNA sequencing and mRNA sequencing data from one of the previously published papers and draw some new conclusions, extending this analysis.

Strengths:

• The authors provide a thorough description of their methods, and a marked-up version of a published protocol that describes how they adapted the protocol to their lab conditions. It should be easy to replicate the experiments.

• The authors test source of bacteria, growth temperature (of both *C. elegans* and bacteria), and light/dark husbandry conditions. They also supply all their raw data, so that sample size for each testing plate can be easily seen (in the supplementary data). None of these variations appears to have a measurable effect on pathogen avoidance in the F2 generation, with all but one of the experiments failing to exhibit learned pathogen avoidance.

• The small RNA seq and mRNA seq analysis is well performed and extends the results shown in the original paper. The original paper did not give many details of the small RNA analysis, which was an oversight. Although not a major focus of this paper, it is a worthwhile extension on the previous work.

• It is rare that negative results such as these are accessible. Although the authors were unable to determine the reason that their results differ from those previously published, it is important to document these attempts in detail, as has been done here. Behavioral assays are notoriously difficult to perform and public discourse around these attempts may give clarity to the difficulties faced by a controversial field.

Weaknesses:

• Although the "standard" conditions have been tested over multiple biological replicates, many of the potential confounders that may have altered the results have been tested only once or twice. For example, changing the incubation temperature to 25{degree sign}C was tested in only two biological replicates (Exp 5.1 and 5.2) - and one of these experiments actually resulted in apparent pathogen avoidance inheritance in the F2 generation (but not in the F1). An alternative pathogen source was tested in only one biological replicate (Exp 3). Given the variability observed in the F2 generation, increasing biological replicates would have added to the strengths of the report.

• A key difference between the methods used here and those published previously, is an increase in the age of the animals used for training - from mostly L4 to mostly young adults. I was unable to find a clear example of an experiment when these two conditions were compared, although the authors state that it made no difference to their results.

• The original paper reports a transgenerational avoidance effect up to the F5 generation. Although in this work the authors failed to see avoidance in the F2 generation, it would have been prudent to extend their tests for more generations in at least a couple of their experiments to ensure that the F2 generation was not an aberration (although this reviewer acknowledges that this seems unlikely to be the case).

---

## [Author Response]

The following is the authors’ response to the original reviews.

**Public Reviews:**

**Reviewer #1 (Public Review):**
[…] Overall, this is an important paper that demonstrates that one model for transgenerational inheritance in C. elegans is not reproducible. This is important because it is not clear how many of the reported models of transgenerational inheritance reported in *C. elegans* are reproducible. The authors do demonstrate a memory for F1 embryos that could be a maternal effect, and the authors confirm that this is mediated by a systemic small RNA response. There are several points in the manuscript where a more positive tone might be helpful.

We would like to correct the statement made in the second to last sentence. The demonstration of an F1 response to PA14 was first reported by Moore et al., (2019) and then by Pereira et al., (2020) using a different behavioral assay. We merely confirmed these results in our hands, and confirmed the observation, first reported by Kaletsky et al., (2020), that sid-1 and sid-2 are required for this F1 response; although we did find that sid-1 and sid-2 are not required for the PA14-induced increase in daf-7p::gfp expression in ASI neurons in the F1 progeny of trained adults, which had not been addressed in the published work.

Yes, the intergenerational F1 response could be a maternal effect, but the in utero F1 embryos and their precursor germ cells were directly exposed to PA14 metabolites and toxins (non-maternal effect) as well as any parental response, whether mediated by small RNAs, prions, hormones, or other unknown information carriers. While the F1 aversion response does require sid-1 and sid-2, we would not presume that the substrate is therefore an RNA molecule, particularly because the systemic RNAi response supported by sid-1 and sid-2 is via long double-stranded RNA. To date, no evidence suggests that either protein transports small RNAs, particularly single-stranded RNAs.

Strengths:The authors note that the high copy number daf-7::GFP transgene used by the Murphy group displayed variable expression and evidence for somatic silencing or transgene breakdown in the Hunter lab, as confirmed by the Murphy group. The authors nicely use single copy daf-7::GFP to show that neuronal daf-7::GFP is elevated in F1 but not F2 progeny with regards to the memory of PA14 avoidance, speaking to an intergenerational phenotype.The authors nicely confirm that sid-1 and sid-2 are generally required for intergenerational avoidance of F1 embryos of moms exposed to PA14. However, these small RNA proteins did not affect daf-7::GFP elevation in the F1 progeny. This result is unexpected given previous reports that single copy daf-7::GFP is not elevated in F1 progeny of sid mutants. Because the Murphy group reported that daf-7 mutation abolishes avoidance for F1 progeny, this means that the sid genes function downstream of daf-7 or in parallel, rather than upstream as previously suggested.

The published report (Moore et al., 2019) shows only multicopy daf-7p::gfp results and does not address the daf-7p::gfp response in sid-1 or sid-2 mutants. Thus, our discovery that systemic RNAi, exogenous RNAi, and heritable RNAi mutants don’t disrupt elevated daf-7p::gfp in ASI neurons in the F1 progeny of PA14 trained P0’s is only unexpected with respect to the published models (Moore et al., 2019, Kaletsky et al., 2020).

The authors studied antisense small RNAs that change in Murphy data sets, identifying 116 mRNAs that might be regulated by sRNAs in response to PA14. Importantly, the authors show that the maco-1 gene, putatively targeted by piRNAs according to the Kaletsky 2020 paper, displays few siRNAs that change in response to PA14. The authors conclude that the P11 ncRNA of PA14, which was proposed to promote interkingdom RNA communication by the Murphy group, is unlikely to affect maco-1 expression by generating sRNAs that target maco-1 in *C. elegans*. The authors define 8 genes based on their analysis of sRNAs and mRNAs that might promote resistance to PA14, but they do not further characterize these genes' role in pathogen avoidance. The Murphy group might wish to consider following up on these genes and their possible relationship with P11.Weaknesses:This very thorough and interesting manuscript is at times pugnacious.

We reiterate that we never claimed that Moore et al., (2019) did not obtain their reported results. We simply stated that we could not replicate their results using the published methods and then failed in our search to identify variable(s) that might account for our results. In revising the manuscript, we have striven to make clear, unmuddied statements of facts and state that future investigations may provide independent evidence that supports the original claims and explains our divergent results.

Please explain more clearly what is High Growth media for *E. coli* in the text and methods, conveying why it was used by the Murphy lab, and if Normal Growth or High Growth is better for intergenerational heritability assays.

We added the standard recipes and the following explanations in the methods section to the revised text.

“NG plates minimally support OP50 growth, resulting in a thin lawn that facilitates visualization of larvae and embryos. HG plates (8X more peptone) support much higher OP50 growth, resulting in a thick bacterial lawn that supports larger worm populations.”

We have also included the following text in our presentation and discussion of the effects of growth conditions on worm choice in PA14 vs OP50 choice assays.

“Furthermore, because OP50 pathogenicity is enhanced by increased *E. coli* nutritive conditions (Garsin et al., 2003, Shi et al., 2006), the growth of F1-F4 progeny on High Growth (HG) plates (Moore et al., 2019; 2021b), which contain 8X more peptone than NG plates and therefore support much higher OP50 growth levels, immediately prior to the F1-F4 choice assays may further contribute to OP50 aversion among the control animals.”

We don’t know enough to claim that HG or NG media is better than the other for intergenerational assays, but they are different. Thus, switching between the two in a multigenerational experiment likely introduces unknown variability.

**Reviewer #2 (Public Review):**
This paper examines the reproducibility of results reported by the Murphy lab regarding transgenerational inheritance of a learned avoidance behavior in *C. elegans*. It has been well established by multiple labs that worms can learn to avoid the pathogen pseudomonas aeruginosa (PA14) after a single exposure. The Murphy lab has reported that learned avoidance is transmittable to 4 generations and dependent on a small RNA expressed by PA14 that elicits the transgenerational silencing of a gene in *C. elegans*. The Hunter lab now reports that although they can reproduce inheritance of the learned behavior by the first generation (F1), they cannot reproduce inheritance in subsequent generations.This is an important study that will be useful for the community. Although they fail to identify a "smoking gun", the study examines several possible sources for the discrepancy, and their findings will be useful to others interested in using these assays. The preference assay appears to work in their hands in as much as they are able to detect the learned behavior in the P0 and F1 generations, suggesting that the failure to reproduce the transgenerational effect is not due to trivial mistakes in the protocol. An obvious reason, however, to account for the differing results is that the culture conditions used by the authors are not permissive for the expression of the small RNA by PA14 that the MUrphy lab identified as required for transgenerational inheritance. It would seem prudent for the authors to determine whether this small RNA is present in their cultures, or at least acknowledge this possibility.

We thank the reviewer for raising this issue and have added the following statement to this effect in the revised manuscript.

“We note that previous bacterial RNA sequence analysis identified a small non-coding RNA called P11 whose expression correlates with bacterial growth conditions that induce heritable avoidance (Kaletsky et al., 2020). Critically, *C. elegans* trained on a PA14 ΔP11 strain (which lacks this small RNA) still learn to avoid PA14, but their F1 and F2-F4 progeny fail to show an intergenerational or transgenerational response (Figure 3L in Kaletsky et al., 2020). The fact that we observed an intergenerational (F1) avoidance response is evidence that our PA14 growth conditions induce P11 expression.”

We believe that this addresses the concern raised here.

The authors should also note that their protocol was significantly different from the Murphy protocol (see comments below) and therefore it remains possible that protocol differences cumulatively account for the different results.

As suggested below, we have added to the supplemental documents the protocol we followed for the aversion assay. In our view, this document shows that our adjustments to the core protocol were minor. Furthermore, where possible, these adjustments were explicitly tested in side-by-side experiments for both the aversion assay and the *daf-7p::gfp* expression assay and presented in the manuscript.

To discover the source(s) of discrepancy between our results and the published results we subsequently introduced variations to this core protocol to exclude likely variables (worm and bacteria growth temperatures, assay conditions, worm handling methods, bacterial culture and storage conditions, and some minor developmental timing issues). Again, where possible, the effect of variations was tested in side-by-side experiments for both the aversion assay and the *daf-7p::gfp* expression assay and were presented in or have now been added to the manuscript.

It remains possible that we misunderstood the published Murphy lab protocols, but we were highly motivated to replicate the results so we could use these assays to investigate the reported RNAi-pathway dependent steps, thus we read every published version with extreme care.

**Reviewer #3 (Public Review):**
[…] Strengths:(1) The authors provide a thorough description of their methods, and a marked-up version of a published protocol that describes how they adapted the protocol to their lab conditions. It should be easy to replicate the experiments.

As noted above in response to a suggestion by reviewer #2, we have replaced the annotated published protocol with the protocol that we followed. This will aid other groups' attempts to replicate our experimental conditions.

(2) The authors test the source of bacteria, growth temperature (of both *C. elegans* and bacteria), and light/dark husbandry conditions. They also supply all their raw data, so that the sample size for each testing plate can be easily seen (in the supplementary data). None of these variations appears to have a measurable effect on pathogen avoidance in the F2 generation, with all but one of the experiments failing to exhibit learned pathogen avoidance.

We note that the parallel analysis of daf-7p::gfp expression in ASI neurons was also tested for several of these conditions and also failed to replicate the published findings.

(3) The small RNA seq and mRNA seq analysis is well performed and extends the results shown in the original paper. The original paper did not give many details of the small RNA analysis, which was an oversight. Although not a major focus of this paper, it is a worthwhile extension of the previous work.(4) It is rare that negative results such as these are accessible. Although the authors were unable to determine the reason that their results differ from those previously published, it is important to document these attempts in detail, as has been done here. Behavioral assays are notoriously difficult to perform and public discourse around these attempts may give clarity to the difficulties faced by a controversial field.

Thank you for your support. Choosing to pursue publication of these negative results was not an easy decision, and we thank members of the community for their support and encouragement.

Weaknesses:(1) Although the "standard" conditions have been tested over multiple biological replicates, many of the potential confounders that may have altered the results have been tested only once or twice. For example, changing the incubation temperature to 25{degree sign}C was tested in only two biological replicates (Exp 5.1 and 5.2) - and one of these experiments actually resulted in apparent pathogen avoidance inheritance in the F2 generation (but not in the F1). An alternative pathogen source was tested in only one biological replicate (Exp 3). Given the variability observed in the F2 generation, increasing biological replicates would have added to the strengths of the report.

We agree that our study was not exhaustive in our exploration of variables that might be interfering with our ability to detect F2 avoidance. We also note that some of these variables also failed (with many more independent experiments) to induce elevated daf-7p::gfp expression in ASI neurons in F2 progeny. Our goal was not to show that variation in some growth or assay condition would generate reproducible negative results, but the exploration was designed to tweak conditions to enable detection of a robust F2 response. Given the strength of the data presented in Moore et al., (2019) we expected that adjustment of the problematic variable would produce positive results apparent in a single replicate, which could then be followed up. If we had succeeded, then we would have documented the conditions that enabled robust F2 inheritance and would have explored molecular mechanisms that support this important but mysterious process.

(2) A key difference between the methods used here and those published previously, is an increase in the age of the animals used for training - from mostly L4 to mostly young adults. I was unable to find a clear example of an experiment when these two conditions were compared, although the authors state that it made no difference to their results.

We can state firmly that the apparent time delay did not affect P0 learned avoidance (new Figure S1) or, as documented in Table S1, daf-7p::gfp expression in ASI neurons. In our experience, training mostly L4’s on PA14 frequently failed to produce sufficient F1 embryos for both F1 avoidance assays or daf-7p::gfp measurements in ASI neurons and collection of F2 progeny. Indeed, in early attempts to detect heritable PA14 aversion, trained P0 and F1 progeny were not assayed in order to obtain sufficient F2’s for a choice assay. These animals failed to display aversion, but without evidence of successful P0 training or an F1 intergenerational response this was deemed a non-fruitful trouble-shooting approach. We have added supplemental Figure S1 which presents P0 choice assay results from experiments using younger trained animals that failed to produce sufficient F1’s to continue the inheritance experiments.

The different timing at the start of training between the two protocols may reflect the age of the recovered bleached P0 embryos. It is reasonable to assume that bleaching day 1 adults vs day 2 or 3 adults from the P-1 population could shift the average age of recovered P0 embryos by several hours. The Murphy protocol only states that P0 embryos were obtained by bleaching healthy adults. Regardless, if the hypothesis entertained here is true, that a several hour difference in larval/adult age during 24 hours of training affects F2 inheritance of learned aversion but does not affect P0 learned avoidance, then we would argue that this paradigm for heritable learned avoidance, as described in Moore et al., (2019, 2021), is not sufficiently robust for mechanistic investigations.

(3) The original paper reports a transgenerational avoidance effect up to the F5 generation. Although in this work the authors failed to see avoidance in the F2 generation, it would have been prudent to extend their tests for more generations in at least a couple of their experiments to ensure that the F2 generation was not an aberration (although this reviewer acknowledges that this seems unlikely to be the case).

We would point out that we also failed to robustly replicate the F2 response in the *daf-7p::gfp* expression assays. An F2-specific aberration that affects two different assays seems quite unlikely, and it remains unclear how we would interpret a positive result in F3 and F4 generations without a positive result in the F2 generation. Were we to further extend these investigations, we believe that exploration of additional culture conditions would warrant higher priority than extension of our results to the F3 and F4 generations.

**Reviewing Editor Comments:**
The reviewers' suggestions for improving the manuscript were mostly minor, to change the wording in some places and to add some more explanation regarding the methods.What should be highlighted in the section on OP50 growth conditions is that the initial preference for PA14 in the Murphy lab has also been observed by multiple other labs (Bargmann, Kim, Zhang, Abbalay). The fact that this preference was not observed by the Hunter lab is one of several indicators of subtle differences in the environment that might add up to explain the differences in results.

We agree that subtle known and unknown differences in OP50 and PA14 culture conditions can have measurable effects on the detection of PA14 attraction/aversion relative to OP50 attraction/aversion that could obscure or create the appearance of heritable effects between generations. We have added (see below) to the text a fuller description of the variability in the initial or naive preference observed in different laboratories using similar or variant 2-choice assays and culture conditions. It is worth emphasizing that direct comparison of the OP50 growth conditions specified in Moore et al., (2021) frequently revealed a much larger effect on the naïve choice index than is reported between labs (Figure 4).

“Naïve (OP50 grown) worms often show a bias towards PA14 in choice assays (Zhang et al., 2005; Ha et al., 2010; Moore et al., 2019; Pereira et al., 2020; Lalsiamthara and Aballay, 2022). This response, rather than representing an innate attraction to PA14, likely reflects the context of the worm's recent growth on OP50, a mild *C. elegans* pathogen (Garigan et al., 2002; Garsin et al., 2003; Shi et al., 2006). Thus, the naïve worms presented with a choice between a recently experienced mild pathogen (OP50) and a novel food choice (PA14) initially choose the novel food instead of the known mild pathogen (OP50 aversion).

In line with our results, some other groups have also reported higher naïve choice index scores (Lee et al., 2017). This variability in naïve choice may reflect differences in growth conditions of either the OP50 or PA14 bacteria. In addition, we note that among the studies that show naïve worm attraction to *Pseudomonas* (OP50 aversion) there are extensive methodological differences from the methods in Moore et al., (2019; 2021b), including differences in bacterial growth temperature, incubation time, whether the bacteria is diluted or concentrated prior to placement on the choice plates, the concentration of peptone in the choice plates, the length of the choice assay, and the inclusion of sodium azide in the choice assays (Zhang et al., 2005; Ha et al., 2010; Moore et al., 2019; Pereira et al 2020; Lalsiamthara and Aballay, 2022). Thus, the cause of the variability across published reports is not clear.”

Overall, an emphasis on the absence of robustness of the reported results, rather than failure to reproduce them (which can always have many reasons), is appropriate.

We agree that an emphasis on robustness is appropriate and have modified the text throughout the manuscript to shift the emphasis to absence of robustness. This includes a change to the manuscript title, which is now, “Reported transgenerational responses to *Pseudomonas aeruginosa* in *C. elegans* are not robust”

A significant experimental addition would be some attempts to determine whether the bacterial PA14 pathogen in the authors' lab produces the P11 small RNA, which has been proposed to have a causal role in initiating the previously reported transgenerational inheritance.

We acknowledge in the revised manuscript that a subsequent publication (Kaletsky et al., 2020) identified a correlation between PA14 training conditions that induced transgenerational memory and the expression of P11, a *P. aeruginosa* small non-coding RNA (see our response above to Reviewer #2’s similar query). While testing for the presence of P11 in Harvard culture conditions would be an important assay in any study whose purpose was to investigate the proposed P11-mediated mechanism underlying the transgenerational responses reported by the Murphy Lab, our goal was rather to replicate the robust transgenerational (F2) responses to PA14 training and then to investigate in more detail how *sid-1* and *sid-2* contribute to transgenerational epigenetic inheritance. Neither *sid-1* nor *sid-2* are predicted to transport small RNAs or single-stranded RNAs, thus testing for the presence of P11 is less relevant to our goals. Regardless, we note that Figure 3L in Kaletsky et al., (2020) showed that PA14 ΔP11 bacteria failed to induce an F1 avoidance response. Thus, the fact that we observed F1 avoidance implies that our culture conditions successfully induced P11 expression.

**Reviewer #1 (Recommendations For The Authors):**
The abstract could be more positive by concluding that 'We conclude that this example of transgenerational inheritance lacks robustness but instead reflects an example of small RNA-mediated intergenerational inheritance.'

As recommended, we have added additional clarifying information to the abstract and moderated the conclusion sentence.

“We did confirm that the dsRNA transport proteins SID-1 and SID-2 are required for the intergenerational (F1) inheritance of pathogen avoidance, but not for the F1 inheritance of elevated *daf-7* expression. Furthermore, our reanalysis of RNA seq data provides additional evidence that this intergenerational inherited PA14 response may be mediated by small RNAs.”

“We conclude that this example of transgenerational inheritance lacks robustness, confirm that the intergenerational avoidance response, but not the elevated *daf-7p::gfp* expression in F1 progeny, requires *sid-1* and *sid-2*, and identify candidate siRNAs and target genes that may mediate this intergenerational response.”

Differential expression of sRNAs or mRNAs might be better understood quantitatively by presenting data in scatterplots (Reed and Montgomery 2020) rather than in volcano plots.

We agree and have modified Figure 6A and 6B.

This statement in the main text might be unnecessary, as it affects the tenor of the conclusion of this significant manuscript. 'We note that none of the raw data for the published figures and unpublished replicate experiments . . . this hampered our ability to fully compare'.

We have rewritten this paragraph to focus on our goal: to identify the source of the discrepancy between our results and the published results. We considered discarding this statement but ultimately decided that our inability to directly compare our data to that of previously published work is a shortcoming of our study that deserves to be acknowledged and explained.

“Ideally, we would have compared our results with the published results (Moore et al., 2019), to possibly identify additional experimental parameters for further investigation; for example, a quantitative comparison of naïve choice in the P0 and F1 generations could help to determine the role of bacterial growth in the choice assay response. However, none of the raw data for the published figures and unpublished replicate experiments (Moore et al., 2019) were available on the publisher’s website or provided upon request to the corresponding author. In the absence of a quantitative comparison, it remains possible that an explanation for the discrepancies between our results and those of Moore et al., (2019) has been overlooked.”

The final sentence of the Discussion could be tempered and more positive by stating 'Thus independent reproducibility is of paramount concern, and we have tried to be completely transparent as a model for how heritability research should be conducted within the *C. elegans* community'.

Thank you. The suggested sentence nicely captures our intention. We now use it, almost verbatim, as our final sentence.

“Thus, independent reproducibility is of paramount concern, and we have tried to be completely transparent as a model for how heritability research should be presented within the *C. elegans* community.”

**Reviewer #2 (Recommendations For The Authors):**
Specific comments:(1) Protocol: It is difficult to assess from the Methods the exact protocol used by the authors to assay food preference. The annotated Murphy protocol is not sufficient. The authors should provide their own protocol - a detailed lab-ready protocol where every step is outlined, and any steps that deviate from the Murphy lab protocol are called out.

Thank you for this excellent suggestion. We now include a protocol that documents the precise steps, timings, and controls that we followed (S1_aversion_protocol). We also include footnotes to both explain the reasons behind particular steps and to document known differences to the published protocol. Given the thoroughness of this suggested approach, we have thus removed the annotated version of Moore et al., (2021) from the revised submission.

(2) The authors imply in the methods that, unlike the Murphy lab, they did NOT use azide in the assay, and instead used 4oC to "freeze" the worms in place - It is not clear whether this method was used throughout all their assays and whether this could be a source of the difference. This change is NOT indicated in the annotated Murphy lab STAR Protocol they provide in the supplement.

We apologize for the lack of clarity. Concerned that azide may be interfering with our ability to detect heritable silencing we tested and then used cold-induced rigor to preserve worm choice in some choice assay results. This was not a change to the core protocol, but a variation used in some assays to determine whether azide could reduce our ability to detect heritable behavioral responses to PA14 exposure. As Moore et al., (2021) show, too much azide can affect measurement of worm choice. Too little or ineffective azide also can affect measurement of worm choice. Azide also affects bacteria (both OP50 and PA14), which could affect the production of molecules that attract or repel worms, much like performing the assay in light vs dark conditions can influence the measured choice index.

In our hands, cold-induced rigor worked well and within biological replicates was indistinguishable from azide (Figure S10). Thus, we include those results in our analysis and now indicate in Tables 2 and S2 and in Figures 1 and 3 which experiments used which method. As suggested, we now provide a detailed protocol that includes a note describing our precise method for cold-induced rigor.

Also, the number of worms used in each assay needs to be specified (same or different from Murphy protocol?), and whether any worms were "censored" as in the Murphy protocol, and if so on what basis.

While we published the exact number of worms scored in each assay (on each plate) it is unknown how this might compare to the results published in Moore et al., (2019), as the number of animals in the presented choice assays (either per plate or per choice) were not reported. Details on censoring, when to exclude data, and additional criteria to abandon an in-progress experiment are now detailed in the protocol (S1_aversion_protocol)

(3) Several instances in the text cite changes in the protocol as producing "no meaningful differences" without referring to a specific experiment that supports that statement (for example, line 399 regarding azide).

We now include data and methods comparing azide and cold-induced rigor (Supplemental document S1_aversion_protocol, Supplemental Figure S10), and data showing the P0 choice index for 48-52 hour post-bleach L4/young adults (Supplemental Figure S1), in addition to the previously noted absence of effects due to differences in embryo bleaching protocols (Figures 2, 3 and Tables 1, 2, S1, and S2).

(4) If the authors want to claim the irreproducibility of the Murphy lab results, they should use the exact protocol used by the Murphy lab in its entirety. It is not sufficient to show that individual changes do not affect the outcome, since the protocol they use appears to include SEVERAL changes which could cumulatively affect the results. If the authors do not want to do this, they should at least acknowledge and summarize in their discussion ALL their protocol changes.

We acknowledge these minor differences between the protocols we followed and the published methods but disagree that they invalidate our results. We transparently present the effect of known minimal protocol changes. We also present analysis of possible invalidating variations (number of animals in a choice assay). We emphasize that in our hands both measures of TEI, the choice assay and measurement of daf-7p::gfp in ASI neurons, failed to replicate the published transgenerational results.

If the protocol is sensitive to how animals are counted, whether bleached embryos are mixed gently or vigorously or a few hours difference in age at training, then in our view this TEI paradigm is not robust.

See also our response to reviewer #3’s public reviews above.

(5) The authors acknowledge that "non-obvious growth culture differences" could account for the different results. In this respect, the Murphy lab has proposed that the transgenerational effect requires a small RNA expressed in PA14. The authors should check that this RNA is expressed in the cultures they grow in their lab and use for their experiments. This could potentially identify where the two protocols diverge.

The bacterial culture conditions and worm training procedures described in Moore et al., (2019) successfully produced trained P0 animals that transmitted a PA14 aversion response to their F1 progeny. In a subsequent publication (Kaletsky et al., 2020), the Murphy lab showed a correlation between the culture conditions that induce heritable avoidance and the expression of P11, a *P. aeruginosa* small non-coding RNA. As mentioned above in response to Reviewer #2’s public review and the Reviewing Editor’s comments to authors, the Murphy lab showed that PA14 ΔP11 bacteria fail to induce an F1 avoidance response (Figure 3L in Kaletsky et al., (2020)). Thus, the fact that we observed F1 avoidance implies that our culture conditions successfully induced P11 expression. We believe that this addresses the concern raised here. Furthermore, if P11 is not reliably expressed in pathogenic PA14, then the published model is unlikely to be relevant in a natural environment. Again, we thank the reviewer for raising this issue and have added this information to the revised manuscript (see above response to Reviewer #2’s Public Reviews).

(6) Legend to Figure 1: please clarify which experiments were done with which PA14 isolates especially for A-C. What is the origin of the N2 strain used here?

These details from Tables 2 and S2 have been added to Figure 1 panels A-C and Figure 3. Bristol N2, obtained from the CGC (reference 257), was used for aversion experiments.

(7) Growth conditions: "These young adults produced comparable P0 and F1 results (Figure 1, Figure 2, and Figure 3)." It is not clear from the text what specific figure panels need to be compared to examine the effect of the variables described in the text. Please indicate which figure panels should be compared (lines 70-95).

The information for the *daf-7p::gfp* expression experiments displayed in Figure 1 and Figure 2 is presented in Table 1 and Table S1. The data for P0 aversion training using younger animals is now presented in Figure S1.

**Reviewer #3 (Recommendations For The Authors):**
While overall I found this easy to follow and well-written, I think the clarity of the figures could be improved by incorporating some of the information from S2 into Figure 3. Besides the figure label listing the experiment (Exp1, Exp2, etc) it would be helpful to add pertinent information about the experiment. For example Exp 1.1 (light, 20{degree sign}C), Exp1.2 (dark, 20{degree sign}C), Exp 5 (25{degree sign}C, light), etc.

Thank you for the suggestion. These details from Tables 2 and S2 have been added to Figures 1 A-C, and 3.

**Citations**

Moore, R.S., Kaletsky, R., and Murphy, C.T. (2019). Piwi/PRG-1 Argonaute and TGF-beta Mediate Transgenerational Learned Pathogenic Avoidance. Cell *177*, 1827-1841 e1812.Moore, R.S., Kaletsky, R., and Murphy, C.T. (2021). Protocol for transgenerational learned pathogen avoidance behavior assays in *Caenorhabditis elegans*. STAR Protoc *2,* 100384.Kaletsky, R., Moore, R.S., Vrla, G.D., Parsons, L.R., Gitai, Z., and Murphy, C.T. (2020). *C. elegans* interprets bacterial non-coding RNAs to learn pathogenic avoidance. Nature *586*, 445-451.Pereira, A.G., Gracida, X., Kagias, K., and Zhang, Y. (2020). *C. elegans* aversive olfactory learning generates diverse intergenerational effects. J Neurogenet *34*, 378-388.